# Structure learning in polynomial time: Greedy algorithms, Bregman information, and exponential families

**Goutham Rajendran**
University of Chicago
goutham@uchicago.edu

**Bohdan Kivva**
University of Chicago
bkivva@uchicago.edu

**Ming Gao**
University of Chicago
minggao@uchicago.edu

**Bryon Aragam**
University of Chicago
bryon@chicagobooth.edu

## Abstract

Greedy algorithms have long been a workhorse for learning graphical models, and more broadly for learning statistical models with sparse structure. In the context of learning directed acyclic graphs, greedy algorithms are popular despite their worst-case exponential runtime. In practice, however, they are very efficient. We provide new insight into this phenomenon by studying a general greedy score-based algorithm for learning DAGs. Unlike edge-greedy algorithms such as the popular GES and hill-climbing algorithms, our approach is vertex-greedy and requires at most a polynomial number of score evaluations. We then show how recent polynomial-time algorithms for learning DAG models are a special case of this algorithm, thereby illustrating how these order-based algorithms can be rigorously interpreted as score-based algorithms. This observation suggests new score functions and optimality conditions based on the duality between Bregman divergences and exponential families, which we explore in detail. Explicit sample and computational complexity bounds are derived. Finally, we provide extensive experiments suggesting that this algorithm indeed optimizes the score in a variety of settings.

## 1 Introduction

Learning the structure of a graphical model from data is a notoriously difficult combinatorial problem with numerous applications in machine learning, artificial intelligence, and causal inference as well as scientific disciplines such as genetics, medicine, and physics. Owing to its combinatorial structure, greedy algorithms have proved popular and efficient in practice. For undirected graphical models (e.g. Ising, Gaussian) in particular, strong statistical and computational guarantees exist for a variety of greedy algorithms [27, 28]. These algorithms are based on the now well-known forward-backward greedy algorithm [29, 57], which has been applied to a range of problems beyond graphical models including regression [57], multi-task learning [52], and atomic norm regularization [44].

Historically, the use of the basic forward-backward greedy scheme for learning directed acyclic graphical (DAG) models predates some of this work, dating back to the classical greedy equivalence search [GES, 13] algorithm. Since its introduction, GES has become a gold-standard for learning DAGs, and is known to be asymptotically consistent under certain assumptions such as faithfulness and score consistency [13, 34]. Both of these assumptions are known to hold for certain parametric families [21], however, extending GES to distribution-free settings has proven difficult. Furthermore,

35th Conference on Neural Information Processing Systems (NeurIPS 2021).

although GES is in practice extremely efficient and has been scaled up to large problem sizes [43], it lacks polynomial-time guarantees. An important problem in this direction is the development of provably polynomial-time, consistent algorithms for DAG learning in general settings.

In this paper, we revisit greedy algorithms for learning DAGs with an eye towards these issues. We propose a greedy algorithm for this problem—distinct from GES—and study its computational and statistical properties. In particular, it requires at most a polynomial number of score evaluations and provably recovers the correct DAG for properly chosen score functions. Furthermore, we illustrate its intimate relationship with existing order-based algorithms, providing a link between these existing approaches and classical score-based approaches. Along the way, we will see how the analysis itself suggests a family of score functions based on the Bregman information [5], which are well-defined without specific distributional assumptions.

**Contributions**    At a high-level, our goal is to understand what kind of finite-sample and complexity guarantees can be provided for greedy score-based algorithms in general settings. In doing so, we aim to provide deeper insight into the relationships between existing algorithms. Our main contributions can thus be outlined as follows:

- A generic greedy forward-backward scheme for optimizing score functions defined over DAGs. Unlike existing *edge*-greedy algorithms that greedily add or remove edges, our algorithm is *vertex*-greedy, i.e. it greedily adds vertices in a topological sort.
- We show how several existing order-based algorithms from the literature are special cases of this algorithm, for properly defined score functions. Thus, we bring these approaches back under the umbrella of score-based algorithms.
- We introduce a new family of score functions derived from the Bregman information, and analyze the sample and computational complexity of our greedy algorithm for this family of scores.
- We explore the optimization landscape of the resulting score functions, and provide evidence that not only does our algorithm provably recover the true DAG, it does so by globally optimizing a score function.

The last claim is intriguing: It suggests that it is possible to globally optimize certain Bayesian network scores in polynomial-time. In other words, despite the well-known fact that global optimization of Bayesian networks scoring functions is NP-hard [12, 14], there may be natural assumptions under which these hardness results can be circumvented. This is precisely the case, for example, for undirected graphs: In general, learning Markov random fields is NP-hard [50], but special cases such as Gaussian graphical models [6, 33] and Ising models [8, 31, 55] can be learned efficiently. Nonetheless, we emphasize that these results on global optimization of the score are merely empirical, and a proof of this fact beyond the linear case remains out of reach.

**Previous work**    The literature on BNSL is vast, so we focus this review on related work involving score-based and greedy algorithms. For a broad overview of BNSL algorithms, see the recent survey [24] or the textbooks [42, 49]. The current work is closely related to and inspired by generic greedy algorithms such as [27–29, 44, 52, 57]. Existing greedy algorithms for score-based learning include GES [13], hill climbing [11, 51], and A* search [56]. In contrast to these greedy algorithms are global algorithms that are guaranteed to find a global optimum such as integer programming [16, 17] and dynamic programming [36, 46, 47]. Another family of order-based algorithms dating back to [51] centers around the idea of *order search*—i.e. first searching for a topological sort—from which the DAG structure is easily deduced; see also [3, 4, 9, 45, 54]. Recently, a series of order-based algorithms have led to significant breakthroughs, most notable of which are finite-sample and strong polynomial-time guarantees [10, 19, 22, 23, 38]. It will turn out that many of these algorithms are special cases of the greedy algorithm we propose; we revisit this interesting topic in Section 3.1.

## 2   Background

Let $X = (X_1, \ldots, X_d)$ be a random vector with distribution $\mathcal{D}$. The goal of structure learning is to find a DAG $W = (V, E)$, also called a Bayesian network (BN), for the joint distribution $\mathcal{D}$. Traditionally, there have been two dominant approaches to Bayesian network structure learning

(BNSL): Constraint-based and score-based. In constraint-based algorithms such as the PC [48] and MMPC [53] algorithms, tests of conditional independence are used to identify the structure of a DAG via exploitation of $d$-separation in DAG models. Score-based algorithms such as GES [13] define an objective function over DAGs such as the likelihood or a Bayesian posterior, and seek to optimize this score.

To formalize this, denote the space of DAGs on $d$ nodes by DAG and let $S : \text{DAG} \to \mathbb{R}$ be a score function. Intuitively, $S$ assigns to each DAG $W$ a "score" $S(W)$ that evaluates the fit between $W$ and $\mathcal{D}$. In the sequel, we assume without loss of generality that the goal is to minimize the score:

$$\min_{W \in \text{DAG}} S(W). \tag{1}$$

Although this is an NP-hard combinatorial optimization problem, we can ask whether or not it is possible to design score functions $S$ which can be optimized efficiently, and whose minimizers are close to $W$. In order for this problem to be well-posed, there must be a *unique* $W$ that we seek; namely, $W$ must be identifiable from $\mathcal{D}$. The problem of identifiability will be taken up further in Section 4, where it will be connected to the choice of score function. For now, our primary interest is solving the problem (1).

Regarding score-based learning, we highlight a subtle point: Recovering the true DAG is not necessarily the same as minimizing the score function, for instance, see Example 1 in [30]. Score-based algorithms in general attempt to learn the true model by way of minimizing the score but it's possible that the graph which minimizes the score could be different from the true model. In other words, the score may not always be properly calibrated to the model. This is a well-studied problem, see e.g. [20, 21, 26], and it is a fascinating and important open problem to better understand under what assumptions a score minimizer is also the true DAG in nonparametric settings.

**Exact algorithms** Solving problem (1) exactly ("exact" meaning a genuine *global* minimizer of (1) is returned) is known to be NP-hard [12, 14]. Some of the earliest exact methods for score-based learning relied on the following basic idea [35, 39, 46, 47]: Use dynamic programming to search for optimal sinks in $W$, remove these sinks, and recursively find optimal sinks in the resulting subgraph. In doing so, a topological sort of $W$ can be learned, and from this sort, the optimal DAG can be easily learned. In other words, once the topological sort is known, finding the corresponding DAG is relatively easy. In the sequel, we refer to the problem of finding the topological sort of $W$ as *order search*. Unfortunately, searching for optimal sinks involves computing $d2^{d-1}$ local scores, which is both time and memory intensive.

**Poly-time algorithms** Recently, a new family of algorithms based on applying the idea of order search has led to significant breakthroughs in our understanding of this problem [10, 19, 22, 23, 38]. Most notably, unlike the exact algorithms described above, these algorithms run in polynomial-time. The key distinction between these algorithms and exact algorithms is the clever exploitation of specific distributional (e.g. moments) or structural properties (e.g. linearity) of $\mathcal{D}$, and as a result do not optimize a specific score function. In contrast, exact algorithms apply to *any* score $S$, and do not require any distributional assumptions.

**Motivation** It is tempting to want to draw connections between exact algorithms and poly-time algorithms: After all, they both rely on the same fundamental principle of order search. In this paper, we explore this connection from the perspective of *greedy optimization*. In particular, we will show how existing polynomial-time algorithms are special cases of a generic greedy forward-backward search algorithm for solving (1) under specific choices of $S$, and show how this leads to new insights for this problem. We *do not* prove that this algorithm exactly solves (1) (save for the exceptional case of linear models; see Corollary 3.2), however, we provide empirical evidence to support this idea on a variety of linear and nonlinear models in Section 6. Since the score-based learning problem is NP-hard, this is of course not possible without additional assumptions.

**Notation** Let $n$ be the number of samples we observe. Each sample is a vector of the form $X = (X_1, \ldots, X_d)$ on $d$ variables. In this paper, $W$ is used for DAGs and the vertex set is $[d] = \{1, 2, \ldots, d\}$. Naturally, we match vertex $i$ to the variable $X_i$. We denote the set of parents of a vertex $i$ with $\text{pa}_W(i)$, dropping the subscript when it's clear from context. We will also abuse notation and use $W$ for the adjacency matrix of the graph $W$ as well. Let $W^{(i)}$ denote the $i$th column

---

**Algorithm 1:** Greedy Forward-Backward Search

---
**Input:** Dataset $X$, tolerance parameter $\gamma \geq 0$
**Output:** DAG $W$

1   $W = \emptyset$ // $n$-vertex graph with no edges
2   $T = []$ // The ordering
    // Forward phase
3   **for** $iter = 1$ *to* $d$ **do**
4      $i = \arg\min_{i \notin T} S_i(e_T)$ // Minimize jump in score
5      $W = W[T \to i]$
6      $T.append(i)$
    // Backward phase
7   **for** *edge $e$ in $W$* **do**
8      **if** $S(W^{-e}) - S(W) \leq \gamma$ **then**
9        $W = W^{-e}$ // Delete the edge $e$
10 **return** $W$ // Guaranteed to be a DAG

---

of $W$, whose nonzero entries are precisely at the set of parents of vertex $i$. Let $W[j, k]$ denote the $(j, k)$th entry of the matrix.

## 3   The GFBS algorithm

In this section, we will describe the greedy algorithm in a general framework. In subsequent sections, we will specialize to particular models or scores, as necessary. Throughout, we let $S$ be an arbitrary decomposable score. That is, $S(W) = \sum_{i \leq d} S_i(W^{(i)})$ for functions $S_i$, an example of which would be the least-squares loss. All the score functions we study in the sequel will have this property.

For a set of vertices $T$, let $e_T$ denote the indicator vector of $T$. For an edge $e$ of $W$, denote by $W^{-e}$ the matrix $W$ with the entry corresponding to $e$ zeroed out. For any set of vertices $J$ and vertex $i \notin J$, denote by $W[J \to i]$ the matrix $W$ where the $i$th column $W^{(i)}$ is replaced by the indicator vector of $J$. That is,

$$W^{-e}[j, k] = \begin{cases} W[j, k] & \text{if } (j, k) \neq e, \\ 0 & \text{otherwise,} \end{cases} \qquad W[J \to i][j, k] = \begin{cases} W[j, k] & \text{if } k \neq i, \\ 1 & \text{if } k = i \text{ and } j \in J \\ 0 & \text{if } k = i \text{ and } j \notin J \end{cases}$$

In Algorithm 1, we outline a general framework based on greedy forward-backward search to learn a DAG $W$ by attempting to minimize the score $S(W)$. For now, we focus on the algorithm itself, and defer discussions of its soundness to Sections 4-5. We denote this algorithm by GFBS for short. Crucially, in contrast to traditional greedy algorithms for structure learning, GFBS is *vertex-greedy*: Instead of greedily adding edges to $W$, GFBS greedily adds *vertices* to first build up a topological sort $T$ of $W$. Specifically, Line 4 in the algorithm greedily finds the next vertex $i$ to add to the ordering, by comparing the score changes if we set the parents of $i$ to be the vertices already in the ordering. Conceptually, this step is one of the most important differences from GES which adds edges one at a time. We make this distinction clear in Appendix A.

It is worth emphasizing that the output of GFBS is guaranteed to be a DAG. The backward phase is standard in greedy optimization, e.g. Greedy Equivalence Search (GES), and serves to eliminate unnecessary edges. In practice, in the backward phase, we could also process the edges in batches. As we explore in Section 5, in certain cases, this allows us to prove sample complexity upper bounds.

**Computational complexity**   The running time of GFBS is a polynomial in $d$ and the time needed to compute the scores $S_i(\cdot)$. More specifically, GFBS requires $O(d^2)$ score evaluations (compared to $O(d2^d)$ for exact algorithms). Evidently, a key computational concern is the complexity of evaluating the score in the first place. For many models such as linear, generalized linear, and exponential family models, this computation can be carried out in $\text{poly}(n, d)$ time, which implies that GFBS on the whole runs in polynomial time. For nonparametric models, this computation may no longer be

polynomial-time, but the total number of score evaluations is still $O(d^2)$. In particular, GFBS always enjoys an exponential speedup over exact algorithms.

**Comparison to GES**   In the supplement (see Appendix A), we exhibit linear Gaussian SEMs and illustrate how GES differs from GFBS for the least squares score as well as the traditional Gaussian BIC score. We first examine a folklore model where we show that their outputs sometimes differ. We also exhibit a model where they always differ. The key takeaway is that GFBS really is a distinct algorithm from GES.

### 3.1   Connection to equal variance SEM

An important line of work starting with [22] has shown that the assumption of equal variances in a linear Gaussian SEM [40] leads directly to an efficient, order-based algorithm. A similar idea in the setting of so-called quadratic variance function (QVF) DAGs was explored in [38]. In this section, we show that the equal-variance algorithm of [10], Algorithm 1, is a special case of GFBS.

Define a score function as follows:

$$S_{\text{LS}}(W) = \sum_{i=1}^{d} \mathbb{E} \operatorname{var}(X_i \mid \operatorname{pa}_W(i)). \tag{2}$$

A few comments on this score function are in order:

1. The only assumption needed on $X$ for this score to be well-defined is that $\mathbb{E} X X^T$ is well-defined, i.e. $X_i \in L^2$ for each $i$.

2. When $X$ satisfies a linear structural equation model $X = W^T X + z$, minimizing $S_{\text{LS}}$ is equivalent to minimizing the least-squares loss $\sum_{i=1}^{d}(X_i - \langle W^{(i)}, X \rangle)^2$. Loh and Bühlmann [30] have shown that when $\operatorname{cov}(z) = \sigma^2 I$, the unique global minimizer of the least-squares loss is the so-called *equal variance SEM*.

3. More generally, for nonlinear models, we have

$$S_{\text{LS}}(W) = \min_{g_1, \ldots, g_d \sim W} \sum_{i=1}^{d} \mathbb{E}(X_i - g_i(X))^2, \tag{3}$$

where $g_1, \ldots, g_d \sim W$ indicates that for each $i$, $g_i$ depends only on the variables in $\operatorname{pa}_W(i)$. In other words, the minimum is taken over all functions $g_1, \ldots, g_d$ that respect the dependency structure implied by $W$. In this case, $g_i$ is essentially $\mathbb{E}[X_i \mid \operatorname{pa}_W(i)]$.

4. We can use (3) to define an empirical score in the obvious way given i.i.d. samples. Alternatively, the residual variance $\mathbb{E} \operatorname{var}(X_i \mid \operatorname{pa}_W(i))$ can be replaced with any estimator of the residual variance.

The GFBS algorithm consists of two phases: A forward phase and a backward phase. Our claim is that the forward phase of GFBS is identical to the equal-variance algorithm from [10]:

**Proposition 3.1.** *After the forward phase, the ordering $T$ returned by GFBS (Algorithm 1) is the same as the ordering returned by the top-down equal-variance algorithm from Chen et al. [10].*

**Corollary 3.2.** *Assume the linear SEM $X = W^T X + z$ with $\operatorname{cov}(z) = \sigma^2 I$ under the score function* (2). *Then GFBS returns a global minimizer of the problem* (1).

Proposition 3.1 will immediately follow from a more general statement which we prove in Theorem 4.6. An intriguing question is to what extent this observation extends to *nonlinear* models such as additive noise models: While we do not have a proof, our experiments in Section 6 suggest something along these lines is true.

## 4   Bregman scores and identifiability via Bregman information

Motivated by the connection between GFBS, global optimality, and the least squares loss, in this section we establish a nice connection between the greedy algorithm and exponential families via the well-known duality between Bregman divergences and partition functions in exponential families [5]. This can then be used to prove identifiability and recovery guarantees for GFBS.

**Bregman divergences and information** Let $\phi : \mathbb{R} \to \mathbb{R}$ be a strictly convex, differentiable function. Let $d_\phi(x, y) = \phi(x) - \phi(y) - (x - y)\phi'(y)$ be the Bregman divergence associated with $\phi$ and let $I_\phi(\mathcal{D}) = \mathbb{E}_{x \sim \mathcal{D}}[d_\phi(x, \mu)]$ be the associated Bregman information. The Bregman-divergence is a general notion of distance that generalizes squared Euclidean distance, logistic loss, Itakuro-Saito distance, KL-divergence, Mahalanobis distance and generalized I-divergences, among others [5]. The Bregman-information of a distribution is a measure of randomness of the distribution, that's associated with $\phi$. Among others, it generalizes the variance, the mutual-information and the Jensen-Shannon divergence of Gaussian processes [5]. See Appendix B for a brief review of this material and a basic treatment of Legendre duality, which will be used in the next section.

## 4.1 Bregman score functions, duality, and exponential families

By replacing the least squares loss in (1) with a Bregman divergence $d_\phi$, we obtain the following score function, which we call a *Bregman score*:

$$S_\phi(W) = \sum_i \mathbb{E}_X d_\phi(X_i, \mathbb{E}[X_i \mid \mathrm{pa}_W(i)]) = \sum_i \min_{g_1, \dots, g_d \sim W} \mathbb{E}_X d_\phi(X_i, g_i(\mathrm{pa}_W(i))) \quad (4)$$

Before we study the behaviour of GFBS on Bregman scores, it is worth taking a moment to interpret this score function. To accomplish this, let us define the notion of an exponential random family DAG:

**Definition 4.1.** *A DAG $W$ and a distribution $\mathcal{D}$ define an exponential random family (ERF) DAG if (a) $\mathcal{D}$ is Markov with respect to $W$, and (b) The local conditional probabilities come from an exponential family, i.e. $\mathbb{P}(X_i \mid \mathrm{pa}_W(i)) \sim \mathrm{ERF}(g_i, \psi_i)$, where $\psi_i$ is the log-partition function of an exponential family with mean function $g_i(\mathrm{pa}_W(i))$.*

Since $\mathrm{ERF}(g_i, \psi_i)$ parametrizes a conditional distribution, its mean parameter $g_i$ is a function instead of vector, which explains our choice of notation. By the Markov property, any choice of local exponential family $\mathrm{ERF}(g_i, \psi_i)$ gives a well-defined joint distribution. The following lemma makes explicit the relationship between Bregman scores, exponential family DAGs, and the Bregman information. Let $\phi^*$ denote the Legendre dual of $\phi$.

**Lemma 4.2.** *Let $\phi$ be a strictly convex, differentiable function and let $\psi := \phi^*$. Then*

$$S_\phi(W) = \sum_{i \leq d} \mathbb{E}[I_\phi(X_i \mid \mathrm{pa}_W(i))] = -\sum_{i \leq d} \mathbb{E}_X \log p_{g_i, \psi}(X_i \mid \mathrm{pa}_W(i)) - C(X) \quad (5)$$

*where $C(X)$ depends only on $X$ and not the underlying DAG $W$ and $p_{g_i, \psi}$ is the density of an $\mathrm{ERF}(g_i, \psi)$ model.*

The proof of this lemma, which can be found in Appendix C, follows from the well-known correspondence between Bregman divergences and exponential families, given by the dual map $\phi \mapsto \phi^*$: Given a Bregman divergence $\phi$, there is a corresponding exponential family whose log-partition function is given by $\phi^*$ [5] and vice versa.

Importantly, Lemma 4.2 shows that the Bregman score $S_\phi$ is equivalent to the expected negative log-likelihood of an exponential family DAG whose local conditional probabilities all have the same log-partition function $\psi$. This means that minimizing the Bregman score can be naturally associated to maximizing the expected log likelihood of such a model. Similar observations had also been made and used in prior works on PCA [15], clustering [5] and learning theory [18].

## 4.2 Identifiability via Bregman information

Motivated by the connection between exponential family DAGs with the same local log-partition maps, in this section, we state our main assumption that generalizes the equal variance assumptions from prior works.

First, we will need a mild assumption on $W$ that's of similar flavor to causal minimality, but with respect to the Bregman-information we are looking at. Denote $\mathcal{A}_W(i)$ to be the non-descendants of $i$ in the graph $W$,

**Assumption 4.3.** *For all $i \leq d$ and all subsets $Y \subseteq \mathcal{A}_W(i)$ such that $\mathrm{pa}(i) \not\subseteq Y$, $\mathbb{E}[I_\phi(X_i \mid Y)] > \mathbb{E}[I_\phi(X_i \mid \mathrm{pa}(i))]$.*

This assumption essentially asserts that no edge in $W$ is superfluous with respect to the distribution on $X$. Now, we state our main assumption.

**Assumption 4.4** (Equal Bregman-information upon conditioning). *Assume that for a constant $\tau > 0$,*

$$\mathbb{E}[I_\phi(X_i \mid \mathrm{pa}(i))] = \mathbb{E}_w[I_\phi(X_i \mid \mathrm{pa}(i) = w)] = \tau \text{ for all } i \leq n$$

*where $\mathrm{pa}(i)$ are the parents of $i$ in the underlying DAG $W$.*

**Example 4.5** (Special case of ANMs). *Suppose we are working with an ANM. That is, there is a DAG $W$ such that for all $i \leq d$, $X_i = f_i(\mathrm{pa}(i)) + \epsilon_i$ for some function $f_i$, where $\epsilon_i$ are jointly independent noise variables. Then, the above assumption says that there is a constant $\tau \geq 0$ such that for all $i$, $I_\phi(\epsilon_i) = \tau$. When $\phi(x) = x^2$, this is the well-known equal variance assumption.*

We are now ready to state our main theorem.

**Theorem 4.6.** *Consider a model satisfying Assumption 4.3 and Assumption 4.4. Under the Bregman score $S_\phi(W)$, the GFBS algorithm with tolerance parameter $\gamma = 0$ will output the true model.*

As stated, the theorem holds for the population setting. The case of finite samples is studied in detail in Section 5, where we prove the same result given sufficient samples.

**Corollary 4.7.** *A model satisfying Assumption 4.3 and the Equal Bregman-information Assumption 4.4 is identifiable.*

We defer the proof of the main theorem to the supplement, where we prove it for an even more general class of functionals that subsume the Bregman-information. Here, we make the following remarks regarding this proof.

1. The proof is actually shown for general functionals for which "conditioning drops value". Therefore, we don't need to only work with Bregman-information and we can instead work with many uncertainty measures of distributions that have this property. This is useful, for example, to show that non-Bregman-type models such as the QVF model from [38] are identifiable using our framework. As a result, Theorem 4.6 subsumes several known identifiability results such as EQVAR [10, 40], NPVAR [19], QVF-ODS [38], and GHD [37]. See Appendix D for details.

2. A similar proof could be adapted for other functionals of distributions that measure the randomness or uncertainty of the distribution. One class of examples could be generalized entropies [2] such as the Shannon entropy, Rényi entropy or the Tsallis entropy. We leave this for future work.

**Remark 4.8.** *An important reason why our algorithm is efficient is because in line $4$ of Algorithm 1, we only compute a single score for each vertex not in the ordering so far. This works especially nicely with the Bregman score, precisely because conditioning with respect to more variables only lowers the Bregman information of a variable, as is exploited to prove the theorems above.*

**A natural score function for non-parametric multiplicative models**    We study multiplicative noise models of the form $X_i = f(\mathrm{pa}(i))\epsilon_i$ from the perspective of the framework built so far. Examples of such models include growth models from economics and biology [32]. More specifically, we choose $\phi(x) = -\log x$ for which the Bregman divergence $d_\phi$ is the *Itakuro-Saito* distance commonly used in the Signal and Speech processing community. The associated Bregman score is the *Itakuro-Saito* score given by

$$S_\phi(W) = \sum_{i \leq d} (\mathbb{E} \log \mathbb{E}[X_i \mid \mathrm{pa}(i)] - \mathbb{E}[\log X_i]).$$

Interestingly, the equal Bregman-information assumption reduces purely to an assumption about the noise variables, akin to the equal variance assumption in the case of additive noise models. This suggests that for multiplicative models, the Itakuro-Saito score is naturally motivated from the perspective of identifiability. This gives a new insight into the applicability of score-based learning for multiplicative models, with theoretical foundations in our analysis. For details, see Appendix E.

## 5  Sample complexity

To derive a sample complexity bound for GFBS, we first need to compute the Bregman score $S_\phi$; due to decomposability and (5), this reduces to estimating the Bregman information $I_\phi$. Let the

samples be denoted $(X_1^{(t)}, X_2^{(t)}, \ldots, X_d^{(t)})$ for $t = 1, 2, \ldots, n$. Denote the Bregman information of $X_i$ conditioning on a set $A$ with conditional mean plugged in as (after some calculation)

$$S(X_i \mid A) := \mathbb{E}[I_\phi(X_i \mid A)] = \mathbb{E}\phi(X_i) - \mathbb{E}\phi(\mathbb{E}(X_i \mid A)) \tag{6}$$

for some strictly convex, differentiable function $\phi$. To estimate this quantity, we can first apply nonparametric regression to estimate $f_{iA} := \mathbb{E}(X_i \mid A)$ and then take the sample mean:

$$\widehat{S}(X_i \mid A) = \frac{1}{n} \sum_{t \leq n} \phi(X_i^{(t)}) - \frac{1}{n} \sum_{t \leq n} \phi(\widehat{f}_{iA}(A^{(t)})). \tag{7}$$

To show convergence rate of this estimator, we will need some some regularity conditions on $f_{iA}$ and $\phi$. These assumptions are standard in the nonparametric statistics literature, see e.g., [25, Chapters 1, 3]. First, we recall the definition of the Hölder class of functions:

**Definition 5.1.** *For any $r = (r_1, \cdots, r_d)$, $r_i \in \mathbb{N}$, let $|r| = \sum_i r_i$ and $D^r = \frac{\partial^{|r|}}{\partial x_1^{r_1} \cdots \partial x_d^{r_d}}$. The Hölder class $\Sigma(s, L)$ is the set of functions satisfying*

$$|D^r f(x) - D^r f(y)| \leq L \, |x - y|^{s-r}$$

*for all $r$ such that $|r| \leq s$ and $x, y \in \mathbb{R}^d$.*

**Assumption 5.2.** *Suppose for all $i$ and ancestor sets $A$ of $i$, $f_{iA} \in \Sigma(s, L)$. And suppose $\phi(X_i)$, $\phi(f_{iA})$ and $\phi'(f_{iA})$ all have finite second moments.*

Denote $\mathcal{A}_W(i)$ to be the non-descendants of $i$ in graph $W$, then the following lemma says that we have a uniform estimator for the Bregman score:

**Lemma 5.3.** *Suppose the Bregman score and the conditional expectations satisfy Assumption 5.2. Using the estimator defined in (7) yields*

$$\min_{i \in [d], A \subseteq \mathcal{A}_W(i)} \mathbb{P}\left(\left|\widehat{S}(X_i \mid A) - S(X_i \mid A)\right| \leq t\right) \geq 1 - \frac{\delta_n^2}{t^2}$$

*where $\delta_n^2 = C(n^{\frac{-2s}{2s+d}} + n^{-1})$ for some constant $C$.*

Using this estimator, we can bound the sample complexity of the forward pass of GFBS as follows:

**Theorem 5.4** (Forward phase of GFBS)**.** *Suppose the BN satisfies the identifiability condition in Theorem 4.6 and assumptions in Lemma 5.3, denote the gap*

$$\Delta := \min_{\substack{i \in [d], A \subseteq \mathcal{A}_W(i) \\ \mathrm{pa}(i) \not\subseteq A}} S(X_i \mid A) - \tau > 0$$

*Let the ordering returned by the first phase of GFBS to be $\widehat{\pi} = (\widehat{\pi}_1, \cdots, \widehat{\pi}_d)$. If the sample size*

$$n \gtrsim \left(\frac{d^2}{\Delta^2 \epsilon}\right)^{\frac{2s+d}{2s} \vee 1}$$

*then $\mathbb{P}(\widehat{\pi}$ is a valid ordering$) \geq 1 - \epsilon$.*

The causal minimality Assumption 4.3 is equivalent to stating $\Delta > 0$. Theorem 5.4 shows that $\Delta$ in fact controls the hardness of the estimation, which is the gap between the minimum Bregman information when all parents are conditioned on and when some parents are missing.

In this section, to obtain strong bounds on sample complexity, we modify the backward phase of GFBS to be as follows:

**Definition 5.5.** *Let $\widehat{A}_0 = \emptyset$ and for $j \geq 1$, $\widehat{A}_j = \{\widehat{\pi}_i | i = 1, 2, \ldots j\}$. For each $\widehat{\pi}_{j+1}$, we find its parents from $\widehat{A}_j$ in the following way, estimate $S(X_{\widehat{\pi}_{j+1}} \mid \widehat{A}_j)$ and $S(X_{\widehat{\pi}_{j+1}} \mid \widehat{A}_j \setminus i)$ for $i \in \widehat{A}_j$. Then, set*

$$\widehat{\mathrm{pa}}(\widehat{\pi}_{j+1}) = \widehat{A}_j \setminus \left\{i \in \widehat{A}_j \,\middle|\, |\widehat{S}(X_{\widehat{\pi}_{j+1}} \mid \widehat{A}_j) - \widehat{S}(X_{\widehat{\pi}_{j+1}} \mid \widehat{A}_j \setminus i)| \leq \gamma\right\}. \tag{8}$$

This says that we keep an edge $(i, \widehat{\pi}_{j+1})$ depending on its influence on the local score at the vertex $\widehat{\pi}_{j+1}$. If the influence is low, then we discard that edge. For our analysis to work, we process these low-influence edges in batches grouped according to the vertices they are oriented towards. In contrast, Algorithm 1 did not batch the edges and simply processed them one at a time.

**Theorem 5.6** (Backward phase of GFBS). *Suppose the same conditions and sample size in Theorem 5.4 holds, using the backward phase defined in* (8) *with* $\gamma = \Delta/2$ *guarantees* $\mathbb{P}(\widehat{W} = W) \geq 1 - \epsilon$.

Proofs can be found in Appendix F in the supplement.

## 6 Experiments

We conduct experiments to show the performance of GFBS on optimizing the Bregman score. We compare GFBS with existing score-based DAG learning algorithms: Gobnilp [16], NOTEARS [59], and GDS [40]. The implementation of these algorithms and data generating process are detailed in Appendix H. Although previous works have evaluated the structure learning performance of special cases of GFBS such as equal variances, we also include these comparisons in the appendix for completeness. Also, in Appendix G, we investigate the performance of GFBS on models which violate the identifiability Assumption 4.4.

- *Choice of $\phi$.* To show the generality of the Bregman score (4), we investigate two convex functions to define the score: $\phi_1(x) = x^2$ and $\phi_2(x) = -\log x$. They correspond to sum of residual variances and sum of residual Itakuro-Saito (IS) distances respectively.

- *Graph type.* We generate three types of graphs: Markov chains (MC), Erdös-Rényi (ER) graphs, Scale-Free (SF) graphs with different expected number of edges. We let the expected number of edges scale with $d$, e.g. ER-2 stands for Erdös-Rényi with $2d$ edges.

- *Model type.* We simulate the data as $X_i = f_i(\mathrm{pa}(i)) + Z_i$ or $X_i = f_i(\mathrm{pa}(i)) \times Z_i$ for different $\phi$'s, where $Z_i$ is independently sampled from some distribution such that Assumption 4.4 is satisfied. Then we consider the following forms of the parental functions $f_i$: linear (LIN), sine (SIN), additive Gaussian process (AGP), and non-additive Gaussian process (NGP).

The main objective of these experiments is to evaluate the performance of these algorithms in optimizing the score: For this, it is necessary to compute the globally optimal score as a benchmark, which is computationally intensive. As a result, our experiments are restricted to $d = 5$. We use Gobnilp [16] to compute the global minimizer. The results are shown in Figure 1. As expected, GFBS returns a near-globally optimal solution in most cases when the sample size is large. Due to finite-sample errors, in some cases (notably on the IS score), GFBS returns a slightly higher score due to the backward phase, which allows the score to increase slightly in favour of sparser solutions. At a technical level, the issue is that the score does not distinguish I-maps from minimal I-maps, and this is exacerbated on finite samples. Better regularization and parameter tuning should resolve this, which we leave to future work. Nonetheless, the close alignment between GFBS and the globally optimal score suggest that GFBS—and hence the equal variance algorithm—is truly minimizing the score.

## 7 Discussion

We introduced the generic GFBS (Greedy Forward-Backward Search) algorithm for score-based DAG learning. It enjoys the guarantees of always outputting a DAG and running in time polynomial in the input size and the time required to compute the score function. We also showed statistical and sample complexity bounds for this algorithm for the generic Bregman score. We motivate this score by formally connecting it to the negative log-likelihood for all exponential DAG models, and considering the well-known approximation capabilities of exponential families, we expect that the Bregman score and our theoretical results apply to a wide variety of settings. In particular, the Bregman score generalizes the least squares score. For least-squares score, our sample complexity results unify and match or improve existing results such as [10, 19, 22]. For generic Bregman scores, no sample complexity results were known prior to this work to the best of our knowledge and we provide the first such results.

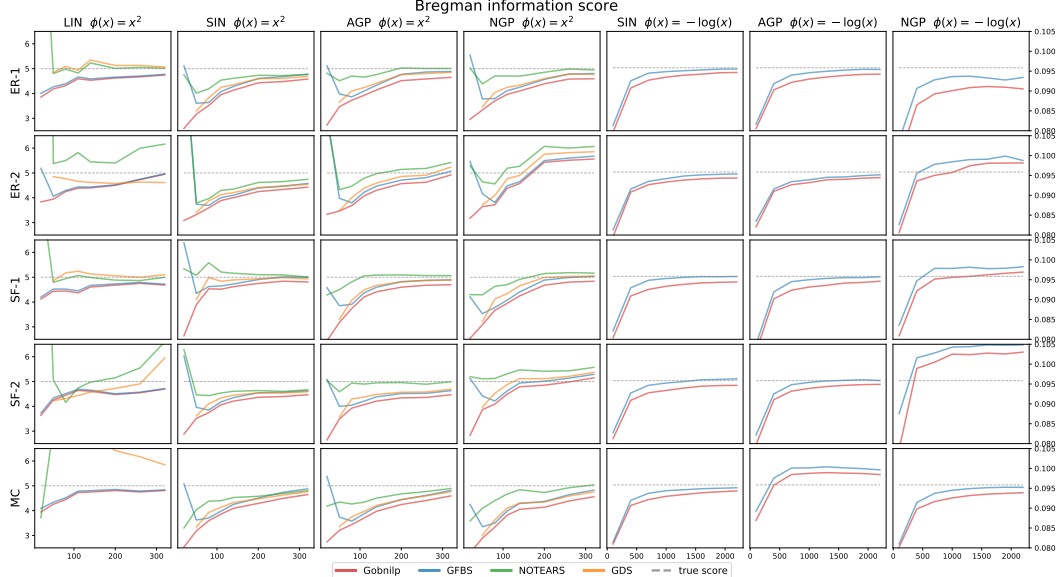

Figure 1: Score of output DAG vs. sample size $n$ for GFBS and 3 other algorithms. Left four columns: $\phi_1(x) = x^2$ and $Z_i$ is $t$-distribution with variance 1; Right three columns: $\phi_2(x) = -\log(x)$ and $Z_i$ is uniform distribution in $[1, 2]$. The two sets of columns have different Y-axis scales. The grey dashed line is the score of the true graph.

The GFBS algorithm also generalizes several prior works on greedy order-based algorithms for DAG learning, e.g., [10, 19, 37, 38]. Existing score-based greedy algorithms (such as GES or hill climbing) are edge-based, whereas these recent order-based algorithms are vertex-based. GFBS shows that each of these prior works can be re-interpreted as score-based greedy algorithms, each of which optimizes a different score. This brings them back under the umbrella of score-based learning. In our statistical guarantees, our assumptions generalize the equal variance assumption that has been studied in the literature in the last decade. Moreover, as a byproduct of our work, we also propose a new score function, the Itakuro-Saito score, for multiplicative SEM models and we leave it to future work to further explore the properties of this score function.

For other future work, it would be insightful to compare Assumption 4.3 to the standard notions such as causal minimality. Moreover, our experiments suggest that the various assumptions we make are not strictly necessary, so an interesting future direction is to study weaker conditions under which GFBS globally optimizes the score.

**Broader impacts**

Learning graphical models has important applications in causal inference, which is useful for mitigating bias in ML models. At the same time, causal models can be easily misinterpreted and provide a false sense of security, especially when they are subject to finite-sample errors. One additional potential negative impact from this line of work is the environmental cost of training large causal models, which can be expensive and time-consuming.

**Acknowledgements**

We thank anonymous reviewers for their helpful comments in improving the manuscript. G.R. was partially supported by NSF grant CCF-1816372. B.K. was partially supported by advisor László Babai's NSF grant CCF 1718902. B.A. was supported by NSF IIS-1956330, NIH R01GM140467, and the Robert H. Topel Faculty Research Fund at the University of Chicago Booth School of Business. All statements made are solely due to the authors and have not been endorsed by the NSF.

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
