# Supplementary Material for "Structure learning in polynomial time: Greedy algorithms, Bregman information, and exponential families"

## A   Comparison to GES

In order to compare GFBS to existing algorithms, in this appendix we present examples to compare the output of GFBS to GES. We first consider a model where this is some ambiguity in the outputs, and then exhibit a model where they always differ. The key takeaway is that GFBS really is a distinct algorithm from GES.

### A.1   A setting where GES sometimes differs from GFBS

We will first consider the following standard example of a non-faithful distribution used in prior works [41] and show how GES differs from GFBS.

Consider a distribution generated as $X_1 = N_1, X_2 = -X_1 + N_2, X_3 = X_1 + X_2 + N_3$ where $N_1, N_2, N_3$ are independent standard Gaussians. We will consider the score function

$$S(W) = \sum_{i \leq 3} (X_i - \sum_{j \in \text{pa}(i)} W_{ji} X_j)^2$$

to be minimized over all matrices $W$ whose support is a DAG.

In the second forward step of GES, there are two equivalence classes that GES could have ended up with because they have the same scores, depending on how the tie is broken. One of them is the graph $X_1 \longrightarrow X_2 \longleftarrow X_3$ and the other is the graph $X_1 \longrightarrow X_2 \longrightarrow X_3$. If GES picked the former and continued with the algorithm, then it ultimately outputs the correct DAG. But if GES picked the latter which it very well could have, then it ends up outputting the wrong DAG $X_1 \longrightarrow X_2 \longleftarrow X_3$ at the end of the algorithm.

On the other hand, as shown in Section 4.1 and Section 5, in both the population setting and the empirical setting for a reasonable sample size, GFBS will provably always output the correct DAG for this distribution since the residual variances are equal.

We also considered the Gaussian BIC score that is traditionally used. In 100 experiments under this score, GES fails all the time (also observed in prior works, for e.g. [41]) and outputs $X_1 \longrightarrow X_2 \longleftarrow X_3$. But we note that GFBS succeeded in all 100 experiments, although we do not give theoretical guarantees for this regularized score.

### A.2   A setting where GES always differs from GFBS

We will tweak the weights of the model from the prior section and show that for this model, GES will always fail whereas GFBS will always succeed for essentially the same reason: Residual variances are equal.

Consider a distribution generated as $X_1 = N_1, X_2 = -X_1 + N_2, X_3 = 0.9X_1 + 0.9X_2 + N_3$. We consider the same score function. We manually verify that GES will always output the DAG $X_1 \longrightarrow X_2 \longleftarrow X_3$ in the population setting. In 100 experiments, GES also always outputted the same wrong DAG. Contrast this to GFBS which will always output the correct DAG in the population setting as well as the empirical setting with a reasonable number of samples.

Finally, under the Gaussian BIC score, in 100 experiments, GES always outputted the wrong DAG and GFBS always outputted the correct DAG, although we do not give theoretical guarantees in general for this phenomenon with the regularized score.

## B   Bregman divergences, Bregman information and Legendre duality

This set of definitions broadly follow the presentation of [5], but is specialized to our setting. Fix a strictly convex, differentiable function $\phi : \mathbb{R} \to \mathbb{R}$.

**Definition B.1.** *Define* $d_\phi : \mathbb{R} \times \mathbb{R} \to \mathbb{R}$ *to be the Bregman-divergence of* $\phi$ *defined as*

$$d_\phi(x, y) = \phi(x) - \phi(y) - (x - y)\phi'(y)$$

*where* $\phi'$ *is the derivative of* $\phi$.

The Bregman-divergence is a general notion of distance that generalizes Squared Euclidean distance, Logistic Loss, Itakuro-Saito distance, KL-divergence, Mahalanobis distance and Generalized I-divergence, among others [5]. In particular, it is nonnegative and is equal to $0$ if and only if the two arguments are equal.

Of particular interest to us, in order to see how it connects to prior works on causal DAG learning, we illustrate with the following example that shows how the variance is a special case of the Bregman-information.

**Example B.2.** *Suppose* $\phi(x) = x^2$. *Then,* $d_\phi(x, y) = (x - y)^2$ *and* $I_\phi(\mathcal{D}) = \mathbb{E}[(x - \mathbb{E}[x])^2] = \mathrm{var}(\mathcal{D})$.

When we study multiplicative DAG models, we study another kind of Bregman-information that arises from the Itakura-Saito distance from signal processing theory. We explore this in more detail in Appendix E.

**Example B.3.** *Assume the domain of the distribution and let* $\phi : \mathbb{R}^+ \to \mathbb{R}$ *be strictly convex.* *Suppose* $\phi(x) = -\log x$. *Then,* $d_\phi(x, y) = \frac{x}{y} - \log \frac{x}{y} - 1$ *and* $I_\phi(\mathcal{D}) = \mathbb{E}[\frac{x}{\mathbb{E}[x]} - \log \frac{x}{\mathbb{E}[x]} - 1] = \log \mathbb{E}[x] - \mathbb{E}[\log x]$.

Any Bregman divergence defines a corresponding *Bregman information*:

**Definition B.4.** *For a distribution* $\mathcal{D}$ *over the reals, define the Bregman-information of* $\phi$ *as*

$$I_\phi(\mathcal{D}) = \mathbb{E}_{x \sim \mathcal{D}}[d_\phi(x, \mu)]$$

*where* $\mu = \mathbb{E}_{x \sim \mathcal{D}}[x]$ *is the mean. For a random variable* $X$ *whose range is distributed as* $\mathcal{D}$ *over* $\mathbb{R}$, *we naturally define* $I_\phi(X) := I_\phi(\mathcal{D})$

The Bregman-information of a distribution is a measure of randomness of the distribution, that's associated with $\phi$. Among others, it generalizes the variance, the mutual-information and the Jensen-Shannon divergence of Gaussian processes [5].

Bregman divergences have the following nice property that we will exploit in our analysis.

**Proposition B.5** ([5, Proposition 1])**.** *The optimization problem*

$$\min_{y \in \mathbb{R}} \mathbb{E}_{x \sim \mathcal{D}}[d_\phi(x, y)]$$

*has a unique minimizer at* $y = \mathbb{E}_{x \sim \mathcal{D}}[x]$.

We note the following:

1. Proposition B.5 is surprising because $d_\phi(x, y)$ is convex with respect to the first argument $x$ but not necessarily with respect to the second argument $y$.

2. Bregman-divergences are the only functionals with this property, i.e. the converse of Proposition B.5 is also true [5, Appendix B]

3. Bregman-information have many other nice properties that make them a useful analytic measure for studying randomness or uncertainty of distributions. See [5] for details.

Now, we briefly review the theory of Legendre duality that will be used in the sequel.

**Definition B.6.** *For a function* $\psi : \mathbb{R} \to \mathbb{R}$, *define the dual function* $\psi^*$ *as*

$$\psi^*(t) = \sup_{\theta \in \mathbb{R}}(t\theta - \psi(\theta))$$

**Proposition B.7.** *For a strictly convex, differentiable function* $\psi : \mathbb{R} \to \mathbb{R}$,

$$\psi^*(t) = tf(t) - \psi(f(t))$$

*where* $f(t) = (\psi')^{-1}(t)$.

*Proof.* Since $\psi$ is strictly convex and differentiable, $\psi'$ is monotonic and hence invertible, so $f$ is well-defined. Now, we can set the derivative of $t\theta - \psi(\theta)$ to zero to obtain that the maximizer $\theta^*$ in Definition B.6 satsifies

$$t = \psi'(\theta^*) \implies \theta^* = f(t)$$

Plugging this back in gives the result. $\qquad\square$

We also note that when $\psi$ is strictly convex and differentiable, $\psi^*$ is also a strictly convex, differentiable function and $(\psi^*)^* = \psi$.

An exponential random family (ERF) is a parametric family of distributions parametrized by the natural parameter $\theta$ with log partition function $\psi$ whose density is given by

$$p_{(\psi,\theta)}(x) = \exp(x\theta - \psi(\theta))p_0(x)$$

The log partition function $\psi$ must be strictly convex and differentiable. This is a general family of distributions that subsumes many standard families of parametric distributions such as the Gaussian distribution, the Poisson distribution and the Bernoulli distribution.

Equivalently, the family could be parameterized by its expectation parameter $\mu = \mathbb{E}_{x \sim p_{(\psi,\theta)}}[x]$

**Fact B.8** ([1, 7]). *For an ERF with natural parameter $\theta$, mean parameter $\mu$, log partition function $\psi$ and dual function $\phi = \psi^*$, we have the following duality:*

$$\mu = \psi'(\theta), \qquad \theta = \phi'(\mu)$$

*Therefore, $\phi'$ and $\psi'$ are inverses of each other.*

For a more general treatment of Legendre duality, see [5].

## C   Proof of Lemma 4.2

When the local conditional probability comes from an exponential family, $\mathbb{P}(X_i | \mathrm{pa}(i)) \sim \mathrm{ERF}(\psi_i, g_i)$ with log-partition function $\psi_i$ and mean function $g_i(\mathrm{pa}(i))$, we can write the density as

$$p_{g_i,\psi_i}(X_i \mid \mathrm{pa}(i)) = p_{(\psi_i,\theta_i)}(X_i \mid \mathrm{pa}(i)) = \exp\left\{X_i\theta_i(\mathrm{pa}(i)) - \psi(\theta_i(\mathrm{pa}(i)))\right\}p_0(X_i) \quad (9)$$

where $\theta_i(\mathrm{pa}(i))$ is the natural parameter corresponding to the mean parameter $g_i(\mathrm{pa}(i))$ associated with an ERF. Note that our notation is consistent with the notation from the previous section.

We need to relate this expression to the Bregman-divergence. Towards that, we have the following lemma.

**Lemma C.1.** *For an ERF with density $p_{(\psi,\theta)}$ with mean parameter $\mu$, we have*

$$d_\phi(x,\mu) = -\log p_{(\psi,\theta)}(x) + \phi(x) + \log p_0(x).$$

A similar result had been obtained in different contexts - PCA [15], clustering [5] and learning theory [18]. A proof follows from essentially similar ideas but we include a proof here for completeness.

*Proof.* Firstly, using Proposition B.7 and Fact B.8, we have

$$\begin{aligned}
\phi(\mu) = \psi^*(\mu) &= \mu((\psi')^{-1}(\mu)) - \psi((\psi')^{-1}(\mu)) \\
&= \mu\phi'(\mu) - \psi(\phi'(\mu)) \\
&= \mu\theta - \psi(\theta).
\end{aligned}$$

We also have

$$-\log p_{(\psi,\theta)}(x) = \psi(\theta) - x\theta - \log p_0(x).$$

Therefore,

$$\begin{aligned}
d_\phi(x,\mu) &= \phi(x) - \phi(\mu) - (x-\mu)\phi'(\mu) \\
&= \phi(x) - \phi(\mu) - (x-\mu)\theta \\
&= \phi(x) - (\mu\theta - \psi(\theta)) - (x-\mu)\theta \\
&= \phi(x) + \psi(\theta) - x\theta \\
&= -\log p_{(\psi,\theta)}(x) + \phi(x) + \log p_0(x). \qquad\square
\end{aligned}$$

Lemma 4.2 now follows immediately.

*Proof of Lemma 4.2.* Using Lemma C.1,

$$
\begin{aligned}
S_\phi(W) &= \sum_i \mathbb{E}_{\mathrm{pa}(i)} I_\phi(X_i | X_{\mathrm{pa}(i)}) \\
&= \sum_i \mathbb{E}_{\mathrm{pa}(i)} \mathbb{E}_{X_i}[d_\phi(X_i, \mathbb{E}[X_i | \mathrm{pa}(i)]) | \mathrm{pa}(i)] \\
&= \sum_i \mathbb{E}_{\mathrm{pa}(i)} \mathbb{E}_{X_i}[d_\phi(X_i, g_i(\mathrm{pa}(i))) | \mathrm{pa}(i)] \\
&= \sum_i \mathbb{E}[d_\phi(X_i, g_i(\mathrm{pa}(i)))] \\
&= \sum_i \mathbb{E}[- \log p_{g_i,\psi}(X_i | \mathrm{pa}(i)) + \phi(X_i) + \log p_0(X_i)] \\
&= - \sum_i \mathbb{E}_X \log p_{g_i,\psi}(X_i | \mathrm{pa}(i)) - C(X)
\end{aligned}
$$

where $C(X)$ depends only on $X$ and not the underlying DAG $W$. □

# D   Proof of Theorem 4.6

We will prove the result for a more general class of functionals $g$ that subsume the Bregman-information.

For a function $f : \mathbb{R} \times \mathbb{R} \longrightarrow \mathbb{R}$ and a distribution $\mathcal{D}$ over $\mathbb{R}$, define $\mu(f, \mathcal{D})$ to be a minimizer (fix an arbitrary choice) of $\mathbb{E}_{x \sim \mathcal{D}}[f(x, y)]$ over $y \in \mathbb{R}$. That is, for all reals $y \in \mathbb{R}$,

$$
\mathbb{E}_{x \sim \mathcal{D}} f(x, y) \geq \mathbb{E}_{x \sim \mathcal{D}}[f(x, \mu(f, \mathcal{D}))]
$$

Fix $f$ and a random variable $X$ with distribution $\mathcal{D}$. Define $g(X) = \mathbb{E}_{X \sim \mathcal{D}}[f(X, \mu(f, \mathcal{D}))]$. The choice of $\mu$, among all possible minimizers, doesn't matter because the value of the functional $g$ will be the same for any such choice. For practical applications, we would generally want $g$ to be efficiently approximable via finite samples.

**Lemma D.1.** *Suppose $f = d_\phi$ for a strictly convex, differentiable $\phi : \mathbb{R} \to \mathbb{R}$. Then, $g$ is the Bregman-information $I_\phi$.*

*Proof.* Using Proposition B.5, we obtain $\mu(f, \mathcal{D}) = \mathbb{E}_{x \sim \mathcal{D}}[x]$. Therefore,

$$
g(X) = \mathbb{E}[f(X, \mu(f, \mathcal{D}))] = \mathbb{E}[d_\phi(X, \mathbb{E}_{X \sim \mathcal{D}}[X])] = I_\phi(X)
$$

□

Consider a distribution $X = (X_1, \ldots, X_d)$ with an underling DAG $W$. Suppose for all $i$,

$$
\mathbb{E}[g(X_i | X_{\mathrm{pa}(i)})] = \mathbb{E}_w[g(X_i | X_{\mathrm{pa}(i)} = w)] = \tau
$$

where $\tau$ is a constant. Note that this reduces to the the equal Bregman-information assumption Assumption 4.4 when $f = d_\phi$.

We now prove the following generalization of a similar result by [19].

**Lemma D.2.** *Let $Y$ be a fixed set of variables. Then, for any $i$ such that $X_i \notin Y$ and no element of $Y$ is a descendant of $X_i$,*

$$
\mathbb{E}[g(X_i | Y)] = \tau \text{ if } X_{\mathrm{pa}(i)} \subseteq Y
$$
$$
\mathbb{E}[g(X_i | Y)] \geq \tau \text{ otherwise}
$$

*Moreover, if for all ancestral sets $Y$ of $i$ such that $\mathrm{pa}(i) \not\subseteq Y$, we had $\mathbb{E}[g(X_i | X_Y)] > \mathbb{E}[g(X_i | X_{\mathrm{pa}(i)})]$, then the inequality above is strict.*

*Proof of Lemma D.2.* Let $\mathcal{D}$ denote the marginal distribution of $X_i$ and let $\mathcal{D}_A$ denote the marginal distribution of $X_i$ conditioned on fixing the variable $A$.

If $X_{\mathrm{pa}(i)} \subseteq Y$, then

$$\mathbb{E}[g(X_i|Y)] = \mathbb{E}[g(X_i|X_{\mathrm{pa}(i)}, Y \setminus X_{\mathrm{pa}(i)})]$$
$$= \mathbb{E}[g(X_i|X_{\mathrm{pa}(i)})]$$
$$= \tau$$

where we used the fact that conditioned on $X_{\mathrm{pa}(i)}$, $X_i$ is independent of $Y \setminus X_{\mathrm{pa}(i)}$.

On the other hand, suppose $X_{\mathrm{pa}(i)} \not\subseteq Y$. Let $Z = X_{\mathrm{pa}(i)} \setminus Y$ be the set of free parent variables. For the sake of brevity, denote by $\mu_Y$ the quantity $\mu(f, \mathcal{D}_Y)$ and denote by $\mu_{Y,Z}$ the quantity $\mu(f, \mathcal{D}_{Y,Z})$. We have

$$\mathbb{E}[g(X_i|Y)] = \mathbb{E}[\mathbb{E}[f(X_i, \mu_Y)|Y]]$$
$$= \mathbb{E}[\mathbb{E}[\mathbb{E}[f(X_i, \mu_Y)|Y, Z]|Y]]$$
$$\geq \mathbb{E}[\mathbb{E}[\mathbb{E}[f(X_i, \mu_{Y,Z})|Y, Z]|Y]]$$
$$= \mathbb{E}[\mathbb{E}[g(X_i|Y, Z)|Y]]$$
$$= \mathbb{E}[g(X_i|Y, Z)]$$
$$= \tau$$

where the inequality followed from the definition of $\mu$ and the last equality used the preceding case that we've already shown, since $X_{\mathrm{pa}(i)} \subseteq Y \cup Z$. Finally, if we had the condition $\mathbb{E}[g(X_i|X_Y)] > \mathbb{E}[g(X_i|X_{\mathrm{pa}(i)})]$ for all ancestral sets $Y$ not containing $pa(i)$, then the inequality in the display above also strictly holds because $Z \neq \emptyset$. $\qquad\square$

We can now prove Theorem 4.6.

*Proof of Theorem 4.6.* Let $f = d_\phi$. Then, we observe that $g = I_\phi$ by Lemma D.1. Therefore, Lemma D.2 can be applied in this setting, and by Assumption 4.3, strict inequality holds.

Consider the forward phase of GFBS. Let the vertices added to $T$ be $v_1, v_2, \ldots, v_d$ respectively in that order. We prove by strong induction on $i$ that for all $i \geq 1$, $v_i$ is a source node (a vertex of indegree 0) of the graph $W \setminus \{v_1, \ldots, v_{i-1}\}$.

To prove the base case, observe that if a vertex $v$ has a parent in $W$, then $\mathbb{E}[g(X_v)] > \mathbb{E}[g(X_v|X_{\mathrm{pa}(v)})] = \tau$. On the other hand, if $v$ is any source node of the graph, then $\mathbb{E}[g(X_v)] = \tau$. Hence, $v_1$ will be a source node of the graph, proving the base case.

Assume the result holds for all indices upto $i$ and consider $v_{i+1}$. Let $H$ be the DAG $W \setminus \{v_1, \ldots, v_i\}$. Consider an arbitrary vertex $w$ in $H$. Firstly, because of the induction hypothesis, no $v_j$ is a descendant of $w$ for any $j \leq i$. Now, if $w$ is a source node in $H$, that is, $\mathrm{pa}(w) \subseteq \{v_1, \ldots, v_i\}$, then $\mathbb{E}[g(X_w|X_{v_1}, \ldots, V_{v_i})] = \mathbb{E}[g(X_w|X_{\mathrm{pa}(w)}] = \tau$, where used the Markov property. On the other hand, if a vertex $w$ in $H$ has a parent in $H$, that is, $\mathrm{pa}(w) \subsetneq \{v_1, \ldots, v_i\}$, then $\mathbb{E}[g(X_w|X_{v_1}, \ldots, V_{v_i})] > \mathbb{E}[g(X_w|X_{\mathrm{pa}(w)}] = \tau$. These two assertions prove that $V_{i+1}$ is a source node of $H$, proving the induction step.

Therefore, for all $i \geq 1$, we must have $\mathrm{pa}(v_i) \subseteq \{v_1, \ldots, v_{i-1}\}$. This proves that $T = v_1 \ldots, v_d$ is a topological sorting of the vertices of the graph. In the backward phase, all edges $e = (i, j)$ not in $W$ will be removed from $W$ because the score will not change after removing $e$ because $j$'s current parents will contain $\mathrm{pa}_W(j)$. Ultimately, the true DAG $W$ remains which will be returned by GFBS. $\qquad\square$

We now explain how Proposition 3.1 follows from Theorem 4.6. By Example B.2, if we take $\phi(x) = x^2$, then the corresponding Bregman information is the variance. In [10], they consider a linear SEM with equal error variances. The assumption of equal error variances is precisely the equal Bregman-information Assumption 4.4 we impose. Now, they iteratively find source nodes for the graph and then condition on them. But by the above inductive proof of Theorem 4.6, this is exactly what happens in the forward phase of GFBS. Therefore, GFBS recovers their algorithm when specialized to equal error-variance linear SEMs.

Other examples of functionals $g$ from previous works include:

- NPVAR [19]: $g(X_i \mid A) = \mathrm{var}(X_i \mid A)$
- QVF-ODS [38]: $g(X_i \mid A) = \mathrm{var}(T_i(X_i) \mid A) - \mathbb{E}(T_i(X_i) \mid A)$ with $\tau = 0$, $T_i$ is a linear transformation that depends on $\mathbb{E}(X_i \mid A)$.
- GHD [37]: $g(X_i \mid A) = ((X_i)_r) - \mathbb{E}f_i^{(r)}(\mathbb{E}(X_i \mid A))$ with $\tau = 0$, where $(a)_r = a(a - 1) \cdots (a - r + 1)$ and $f_i^{(r)}$ is a $r$th factorial constant moments ratio (CMR) function of form

$$f_i^{(r)}(x; a(i), b(i)) = x^r \prod_{k=1}^{p_i} \frac{(a_{ik} + r - 1)_r}{a_{ik}^r} \prod_{\ell=1}^{q_i} \frac{b_{i\ell}^r}{(b_{i\ell} + r - 1)_r}$$

such that

$$\mathbb{E}[(X_i)_r \mid \mathrm{pa}(i)] = f_i^{(r)}\big(\mathbb{E}[X_i \mid \mathrm{pa}(i)]; a(i), b(i)\big)$$

for any integer $r \le \max X_i$.

# E   A natural score function for non-parametric multiplicative models

Consider the multiplicative SEM model

$$X_i = f(X_{\mathrm{pa}(i)})\epsilon_i$$

with an underlying DAG $W$. We will also assume that $\epsilon_i$ is positive with probability 1. Examples of such models include growth models from economics and biology [32].

Let $\phi(x) = -\log x$. Then, the Bregman divergence $d_\phi$ will be the Itakuro-Saito distance used in Signal and Speech processing community. From Corollary 4.7, we get that the model is identifiable under the condition

$$\mathbb{E}[I_\phi(X_i | X_{\mathrm{pa}(i)})] = \text{constant}.$$

But we can compute this explicitly for a multiplicative model. Firstly, note that $\mathbb{E}[X_i | X_{\mathrm{pa}(i)} = w] = \mathbb{E}[f(X_{\mathrm{pa}(i)})\epsilon_i | X_{\mathrm{pa}(i)=w}] = f(X_{\mathrm{pa}(i)})\mathbb{E}[\epsilon_i]$. Using the same calculations as in Example B.3, we get

$$\mathbb{E}[I_\phi(X_i | X_{\mathrm{pa}(i)})] = \mathbb{E}_w[\mathbb{E}[-\log \frac{X_i}{\mathbb{E}[X_i | X_{\mathrm{pa}(i)} = w]} | X_{\mathrm{pa}(i)} = w]]$$

$$= \mathbb{E}_w[\mathbb{E}[-\log \frac{\epsilon_i}{\mathbb{E}[\epsilon_i]} | X_{\mathrm{pa}(i)} = w]]$$

$$= \log \mathbb{E}[\epsilon_i] - \mathbb{E}[\log \epsilon_i]$$

Therefore, the equal Bregman-information assumption is equivalent to the following assumption on the noise variables

$$\log \mathbb{E}[\epsilon_i] - \mathbb{E}[\log \epsilon_i] = \text{constant}$$

This is satisfied for instance when $\epsilon_i$ are identically distributed. Our theory of Bregman scores illustrates that when such assumptions are feasible, such as in the case of identically distributed noise variables, then to estimate such models via score based approaches, a great candidate score would be the Itakuro-Saito score

$$S_\phi(W) = \sum_{i \le d} \mathbb{E}[I_\phi(X_i | X_{\mathrm{pa}(i)})] = \sum_{i \le d} (\mathbb{E} \log \mathbb{E}[X_i | X_{\mathrm{pa}(i)}] - \mathbb{E}[\log X_i]).$$

# F   Proofs for Section 5

## F.1   Proof of Lemma 5.3

*Proof.* For all $i \in [d]$ and $A \subseteq \mathcal{A}_G(i)$,

$$\mathbb{E}\left(\widehat{S}(X_i \mid A) - S(X_i \mid A)\right)^2 \lesssim \mathbb{E}\left(\mathbb{E}\phi(X_i) - \frac{1}{n}\sum_t \phi(X_i^{(t)})\right)^2 + \mathbb{E}\left(\mathbb{E}\phi(f_{iA}) - \frac{1}{n}\sum_t \phi\left(\widehat{f}_{iA}(A^{(t)})\right)\right)^2$$

$$\lesssim n^{-1} + \mathbb{E}\left( \mathbb{E}\phi(f_{iA}) - \frac{1}{n}\sum_t \phi\left( \widehat{f}_{iA}(A^{(t)}) \right) \right)^2$$

due to the finite second moment and parametric rate. For the second term,

$$\mathbb{E}\left( \mathbb{E}\phi(f_{iA}) - \frac{1}{n}\sum_t \phi\left( \widehat{f}_{iA}(A^{(t)}) \right) \right)^2$$

$$= \mathbb{E}\left( \mathbb{E}\phi(f_{iA}) - \frac{1}{n}\sum_t \phi\left( f_{iA}(A^{(t)}) \right) + \frac{1}{n}\sum_t \phi\left( f_{iA}(A^{(t)}) \right) - \frac{1}{n}\sum_t \phi\left( \widehat{f}_{iA}(A^{(t)}) \right) \right)^2$$

$$\lesssim \mathbb{E}\left( \mathbb{E}\phi(f_{iA}) - \frac{1}{n}\sum_t \phi\left( f_{iA}(A^{(t)}) \right) \right)^2 + \frac{1}{n}\sum_t \mathbb{E}\left( \phi\left( f_{iA}(A^{(t)}) \right) - \phi\left( \widehat{f}_{iA}(A^{(t)}) \right) \right)^2$$

$$\lesssim n^{-1} + \frac{1}{n}\sum_t \mathbb{E}\left( \phi'(f_{tA}(A^{(t)})) \right)^2 \mathbb{E}\left( f_{iA}(A^{(t)}) - \widehat{f}_{iA}(A^{(t)}) \right)^2$$

$$\lesssim n^{-1} + n^{\frac{-2s}{2s+d}}$$

For the first term, the inequality is by finite second moment and parametric rate. For second term, apply first order Taylor expansion and absorb the high order estimation error terms into the constant before the inequality. Finally, the tail probability bound follows by Markov's inequality. □

### F.2 Proof of Theorem 5.4

*Proof.* Let $\widehat{A}_0 = \emptyset$ and for $j \geq 1$, $\widehat{A}_j = \{\widehat{\pi}_i | i = 1, 2, \ldots j\}$. Denote the event $\mathcal{E}_j = \{\widehat{\pi}_j$ is a source node of $G[V \setminus \widehat{A}_{j-1}]\}$. Then

$$\mathbb{P}(\widehat{\pi} \text{ is a valid ordering}) = \prod_{j=0}^{d-1} \mathbb{P}(\mathcal{E}_{j+1} \,|\, \mathcal{E}_j)$$

For each term of the product,

$$\mathbb{P}(\mathcal{E}_{j+1} \,|\, \mathcal{E}_j) = \sum_{\substack{A \text{ is a subset of non-descendants} \\ |A|=j}} \mathbb{P}(\mathcal{E}_{j+1} \,|\, \widehat{A}_j = A, \mathcal{E}_j) \, \mathbb{P}(\widehat{A}_j = A \,|\, \mathcal{E}_j)$$

$\mathcal{E}_j$ implies that $\widehat{A}_j = A$ is of size $j$ and a subset of non-descendants of remaining nodes. More importantly, all possibilities sum up to one

$$\sum_{\substack{A \text{ is a subset of non-descendants} \\ |A|=j}} \mathbb{P}(\widehat{A}_j = A \,|\, \mathcal{E}_j) = 1$$

Invoking Lemma 5.3, union bound the estimation error

$$\mathbb{P}\left( \cup_{i \notin V \setminus A} \left\{ |\widehat{S}(X_i \,|\, A) - S(X_i \,|\, A)| \geq t \right\} \right) \leq \sum_{i \notin V \setminus A} \mathbb{P}\left( |\widehat{S}(X_i \,|\, A) - S(X_i \,|\, A)| \geq t \right) \leq (d-j)\frac{\delta_n^2}{t^2}$$

Thus, with probability at least $1 - (d-j)\delta_n^2/t^2$, we have

$$\begin{cases} \widehat{S}(X_i \,|\, A) \leq \tau + t & i \text{ is a source node of } G[V \setminus A] \\ \widehat{S}(X_i \,|\, A) \geq \tau + \Delta - t & i \text{ is not a source node of } G[V \setminus A] \end{cases}$$

Therefore, with $t \leq \Delta/2$, the node $\widehat{\pi}_{j+1}$ found by GFBS which minimizes the score is still a source node. This implies for all possible $A$,

$$\mathbb{P}(\mathcal{E}_{j+1} \,|\, \widehat{A}_j = A, \mathcal{E}_j) \geq 1 - 4(d-j)\frac{\delta_n^2}{\Delta^2}$$

furthermore,

$$\mathbb{P}(\mathcal{E}_{j+1} \mid \mathcal{E}_j) \geq 1 - 4(d-j)\frac{\delta_n^2}{\Delta^2}$$

and finally

$$\mathbb{P}(\widehat{\pi} \text{ is a valid ordering}) = \prod_j \mathbb{P}(\mathcal{E}_{j+1} \mid \mathcal{E}_j) \geq 1 - \sum_{j=0}^{d-1}(d-j)\frac{4\delta_n^2}{\Delta^2} \geq 1 - \frac{4d^2\delta_n^2}{\Delta^2}$$

Solving $\mathbb{P}(\widehat{\pi} \text{ is a valid ordering}) > 1 - \epsilon$ yields the desired result. $\qquad\square$

### F.3   Proof of Theorem 5.6

*Proof.* We only need to show parents of each node are correctly estimated. Theorem 5.4 guarantees that the parents of $\widehat{\pi}_{j+1}$ are in $\widehat{A}_j$. Thus,

$$S(\widehat{\pi}_{j+1} \mid \widehat{A}_j) = S(\widehat{\pi}_{j+1} \mid \mathrm{pa}(\widehat{\pi}_{j+1}))$$

By the definition of $\Delta$,

$$\begin{cases} S(X_{\widehat{\pi}_{j+1}} \mid \widehat{A}_j \setminus i) - S(X_{\widehat{\pi}_{j+1}} \mid \widehat{A}_j) \geq \Delta & i \in \mathrm{pa}(\widehat{\pi}_{j+1}) \\ S(X_{\widehat{\pi}_{j+1}} \mid \widehat{A}_j \setminus i) - S(X_{\widehat{\pi}_{j+1}} \mid \widehat{A}_j) = 0 & i \notin \mathrm{pa}(\widehat{\pi}_{j+1}) \end{cases}$$

Invoking Lemma 5.3, with probability at least $1 - d\delta_n^2/t^2$, for all $i \in \widehat{A}_j \cup \{\emptyset\}$

$$|\widehat{S}(X_{\widehat{\pi}_{j+1}} \mid \widehat{A}_j \setminus i)) - S(X_{\widehat{\pi}_{j+1}} \mid \widehat{A}_j \setminus i))| \leq t$$

which implies

$$\begin{cases} |\widehat{S}(X_{\widehat{\pi}_{j+1}} \mid \widehat{A}_j) - \widehat{S}(X_{\widehat{\pi}_{j+1}} \mid \widehat{A}_j \setminus i)| \leq 2t & i \in \mathrm{pa}(\widehat{\pi}_{j+1}) \\ |\widehat{S}(X_{\widehat{\pi}_{j+1}} \mid \widehat{A}_j) - \widehat{S}(X_{\widehat{\pi}_{j+1}} \mid \widehat{A}_j \setminus i)| \geq \Delta - 2t & i \notin \mathrm{pa}(\widehat{\pi}_{j+1}) \end{cases}$$

With $2t \leq \Delta/2 = \gamma$, we can distinguish parents of $\widehat{\pi}_{j+1}$ from other non-descendants, thus $\widehat{\mathrm{pa}}(\widehat{\pi}_{j+1}) = \mathrm{pa}(\widehat{\pi}_{j+1})$. A union bound over $d$ nodes gives us the same sample complexity as in Theorem 5.4, which completes the proof. $\qquad\square$

## G   Unequal Bregman score cases

In this section, we investigate the behaviour of GFBS when the equal Bregman information condition is violated. As is evident from the proofs in Appendix D, exact equality is actually not necessary for the proof and the algorithm to go through. The Assumption 4.4 has a straightforward extension analogous in the literature [22], which also ensures identifiability. We present the result here, whose proof follows Appendix D and previous work and thus is omitted.

**Assumption G.1.** *There exists a valid ordering $\pi$ such that for all $i \in [d]$ and $\ell \in \pi_{[i+1:d]}$,*

$$\mathbb{E}[I_\phi(X_i \mid X_{\pi_{[1:i-1]}})] = \mathbb{E}[I_\phi(X_i \mid \mathrm{pa}(i))] < \mathbb{E}[I_\phi(X_\ell \mid X_{\pi_{[1:i-1]}})]$$

To demonstrate this assumption, we conduct experiments with $\phi(x) = x^2$, which leads to $\mathbb{E}[I_\phi(X_i \mid \mathrm{pa}(i))] = \mathbb{E}\,\mathrm{var}(X_i \mid \mathrm{pa}(i))$. We can generate data satisfying this "unequal" assumption for Markov chain + sine model + Gaussian noise. The idea is that to restrict the range of noise variance.

Suppose the Markov chain is $X_1 \to \cdots \to X_d$, with $X_i = \sin(X_{i-1}) + Z_i$, and $Z_i \sim \mathcal{N}(0, \sigma_i^2)$. We restrict the $\sigma_i^2$ to be sampled from $[1, 1.2]$. To make sure Assumption G.1 is satisfied, we need for any $i$ and $i < \ell \leq d$,

$$\sigma_i^2 = \mathbb{E}\,\mathrm{var}(X_i \mid X_{i-1}) < \mathbb{E}\,\mathrm{var}(X_\ell \mid X_{i-1}) = \mathbb{E}\,\mathrm{var}(\sin(X_{\ell-1} + \epsilon_{\ell-1}) \mid X_{i-1}) + \sigma_\ell^2$$

It suffices to find a lower bound on $\mathrm{var}(\sin(a + X))$ for any $a \in [-1, 1]$ and $X \sim \mathcal{N}(0, \sigma^2)$.

**Lemma G.2.** *Suppose $X \sim \mathcal{N}(0, \sigma^2)$ with $\sigma^2 \geq 1$, then for any $a \in [-1, 1]$, $\mathrm{var}(\sin(X+a)) \geq 1/4$.*

With Lemma G.2, we can show that the identifiability is guaranteed:

$$\mathbb{E}\operatorname{var}(\sin(X_{\ell-1} + \epsilon_{\ell-1}) \,|\, X_{i-1}) \geq \mathbb{E}(1/4 \,|\, X_{i-1}) = 1/4 > 0.2 = \max(\sigma_i^2 - \sigma_\ell^2)$$

*Proof of Lemma G.2.* Using the identity

$$\sin(X + a) = \sin a \cos X + \cos a \sin X$$

we have

$$\mathbb{E}\sin(X + a) = \sin a \mathbb{E}\cos X = \sin a \times \operatorname{Re}[\mathbb{E}\exp(-iX)] = \sin a \exp(-\sigma^2/2).$$

Moreover, a short calculation shows that

$$\mathbb{E}\sin^2(X + a) = \frac{1}{2} - \frac{1}{2}\cos 2a \exp(-2\sigma^2).$$

Finally,

$$\begin{aligned}
\operatorname{var}\sin(X + a) &= \mathbb{E}\sin^2(X + a) - (\mathbb{E}\sin(X + a))^2 \\
&= \frac{1}{2} - \frac{1}{2}e^{-2\sigma^2} + \sin^2 a \times e^{-\sigma^2}(e^{-\sigma^2} - 1) \\
&> 1/4.
\end{aligned} \qquad \square$$

To illustrate this condition, we run a simple experiment as follows: Consider two settings for $\sigma_i^2$: Sampled uniformly and randomly from (a) $[1, 1.2]$ or (b) $[0.1, 1.9]$. Run GFBS and Gobnilp on generated data, then compare the score obtained by two algorithms, and test whether the ordering of estimated graph is correct. Since the true graph is a Markov chain, there is only one true ordering.

As shown in Figure 2, when Assumption G.1 is satisfied through Lemma G.2, the score output by GFBS is close to the true one, and the topological ordering can be recovered. When the range of $\sigma_i^2$ is not well-controlled, GFBS does not return the correct ordering, and neither does Gobnilp. Interestingly, GFBS nonetheless does a good job at optimizing the score.

## H    Experiment details

In this appendix we collect all the details of the experiments in Section 6.

### H.1    Experiment settings

**Bregman scores**: We define them through convex functions

- *Residual variances*: $\phi_1(x) = x^2$
- *Itakuro-Saito*: $\phi_2(x) = -\log(x)$

**Graph types**: We let the expected number of edges to scale with $d$, e.g. ER-2 stands for Erdös-Rényi with $2d$ edges.

- *ER*: randomly choose $s$ edges from all possible $\binom{d}{2}$ directed edges, then randomly permute the nodes
- *SF*: scale-free graphs generated through Barabasi-Albert process
- *MC*: Markov chain, randomly permute the nodes

**Model types**: We specify the parental functions $f_i$ to be as follows: linear model (LIN), sine model (SIN), additive Gaussian Process (AGP) and non-additive Gaussian Process (NGP) with with kernel $K(x, y) = e^{-|x-y|^2/2}$.

- For $\phi_1$, data is generated according to the form of $X_i = f_i(\operatorname{pa}(i)) + Z_i$
    - $\sigma$: additive noise standard deviation, set to 1
    - Noise distribution:

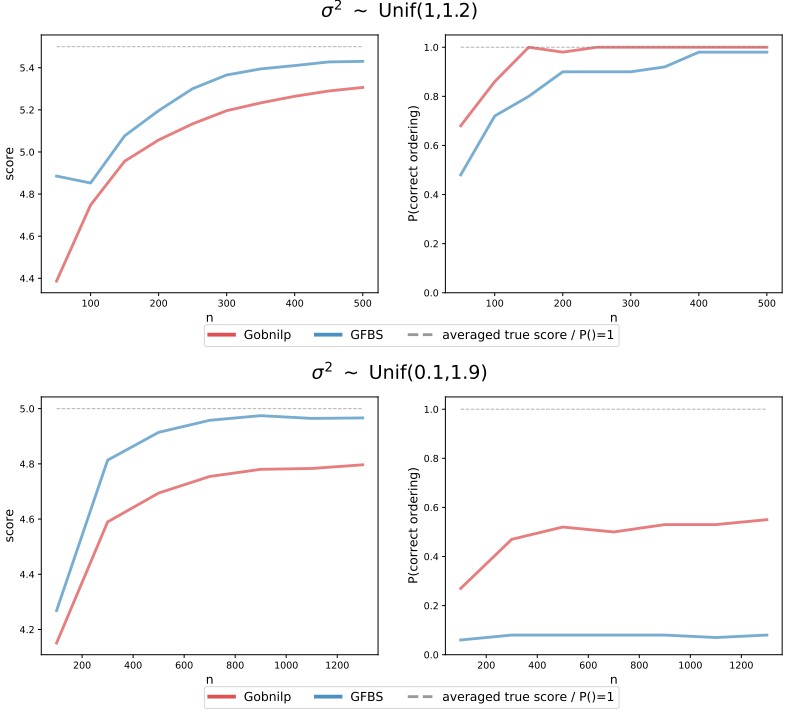

Figure 2: Unequal Bregman score experiments. Left column: score output by GFBS and Gobnilp; Right column: $\mathbb{P}$(correct ordering) v.s. sample size; Upper row: range of $\sigma_i^2$ is $[1, 1.2]$; Bottom row: range of $\sigma_i^2$ is from $[0.1, 1.9]$. The gray dashed lines indicate average true score ($\sum_i \sigma_i^2$, left) or the optimal probability of recovery (right).

- ∗ *Gaussian*: $Z_i = \sigma \times \mathcal{N}(0, 1)$
- ∗ *t*: $Z_i = \sigma \times t(3)/\sqrt{3}$
- ∗ *Gumbel*: $Z_i = \sigma \times Gumbel(0, \sqrt{6}/\pi)$
  - – Parental functions:
    - ∗ *LIN*: $f_i = \sum_{\ell \in \mathrm{pa}(i)} \beta_{i\ell} X_\ell$, where $\beta_{i\ell} = Rademacher \times Unif(0.5, 1.2)$
    - ∗ *SIN*: $f_i = \sum_{\ell \in \mathrm{pa}(i)} \sin(X_\ell)$
    - ∗ *AGP*: $f_i = \sum_{\ell \in \mathrm{pa}(i)} GP(X_\ell)$
    - ∗ *NGP*: $f_i = GP(\mathrm{pa}(i))$
- For $\phi_2$, data is generated according to the form of $X_i = f_i(\mathrm{pa}(i)) \times Z_i$.
  - – Noise distribution:
    - ∗ *Uniform*: $Z_i \sim Unif(1, 2)$
  - – Parental function:
    - ∗ *SIN*: $f_i = \frac{1}{|\mathrm{pa}(i)|} \sum_{\ell \in \mathrm{pa}(i)} \sin^2(X_\ell)$;
    - ∗ *AGP*: $f_i = \frac{1}{|\mathrm{pa}(i)|} \sum_{\ell \in \mathrm{pa}(i)} GP^2(X_\ell) + 0.5$;
    - ∗ *NGP*: $f_i = \frac{1}{2} GP^2(\mathrm{pa}(i)) + 0.5$

**Other parameters:**

- Dimension: $d = 5, 10, 20, 30$
- Number of edges: $s = kd$, $k = 1, 2, 4$
- Sample size: $n = [50, 80, 110, 140, 200, 260, 320]$ for $\phi_1$, $n = [100, 400, 700, ..., 2200]$ for $\phi_2$
- Replications of simulation: $N = 30$

## H.2 Implementation of algorithms

For GFBS, use Generalized Additive Model (GAM) to estimate all conditional expectations and compute all local scores as reference for all other methods. In particular, GAM is replaced by ordinary least square for *LIN* model. In backward phase, use threshold $\gamma = 0.05$ for $\phi_1$ and $\gamma = 0.0005$ for $\phi_2$. GAM is implemented by Python package `pygam` with default parameters to avoid favoring one particular method due to hyper-parameter tuning,

We compare GFBS with following score-based structure learning algorithms:

- Gobnilp[16]: is an exact solver for score-based Baysian network learning through Constraint Integer Programming. We input the local scores output by GFBS for it to optimize. The implementation is available at `https://www.cs.york.ac.uk/aig/sw/gobnilp/`.

- NOTEARS[58, 59]: uses an algebraic characterization of DAGs for score-based structure learning of nonparametric models via partial derivatives. We adopt example hyper-parameters to run, then compute the total score of output DAG using local scores output by GFBS. The implementation is available at `https://github.com/xunzheng/notears`.

- GDS[40]: greedily searches over neighbouring DAGs differed by adding / deleting / reversing one edge. Switch the score from log likelihood to our score setting, use `gam` function in R package `mgcv` with P-splines `bs='ps'` and the default smoothing parameter `sp=0.6` to estimate the conditional expectations. In particular, GAM is replaced by ordinary least square for *LIN* model. Implementation is available at `https://academic.oup.com/biomet/article/101/1/219/2364921#supplementary-data`. Omitted for $d > 10$ due to computational cost.

- GES[13]: greedily searches over neighbouring Markov equivalence class to optimize the score. We use `sem-bic` score with `penaltyDiscount=0`, which amounts to score equals $BIC = 2L$ where $L$ is the likelihood. Only run for linear model and compare SHD. Implementation is available at `https://github.com/bd2kccd/py-causal`.

These simulations used an Intel E5-2680v4 2.4GHz CPU running on an internal cluster.

## H.3 Evaluation metrics

- Structural Hamming Distance (SHD): common metric for comparing performance in structure learning, which counts the total number of edge additions, deletions, and reversals needed to convert the estimated graph into the true graph.

- Bregman Score: $\sum_i \mathbb{E}\phi(X_i) - \mathbb{E}\phi(\mathbb{E}(X_i \mid \mathrm{pa}(i)))$. Except for the true score indicated by grey dashed lines, this metric is evaluated in finite sample using estimator defined in (7).

## H.4 Additional experiments

Here we present some additional experiments.

### H.4.1 Main figures with other noise

In Figure 3, we present the left four columns of Figure 1 under other two noise distribution: Gaussian and Gumbel. Note that the experiment settings are under $\phi_1$.

### H.4.2 Structure learning

We illustrate the performance of GFBS on structural learning by considering experiment settings under higher dimensions and $\phi_1$, where Gobnilp and GDS are omitted due to heavy computational cost.

- Figure 4: SHD v.s. sample size $n$ for $d = 20, 30$, Gaussian noise, and $\phi_1$.
- Figure 5: SHD v.s. sample size $n$ for $d = 20, 30$, t noise, and $\phi_1$.
- Figure 6: SHD v.s. sample size $n$ for $d = 20, 30$, Gumbel noise, and $\phi_1$.

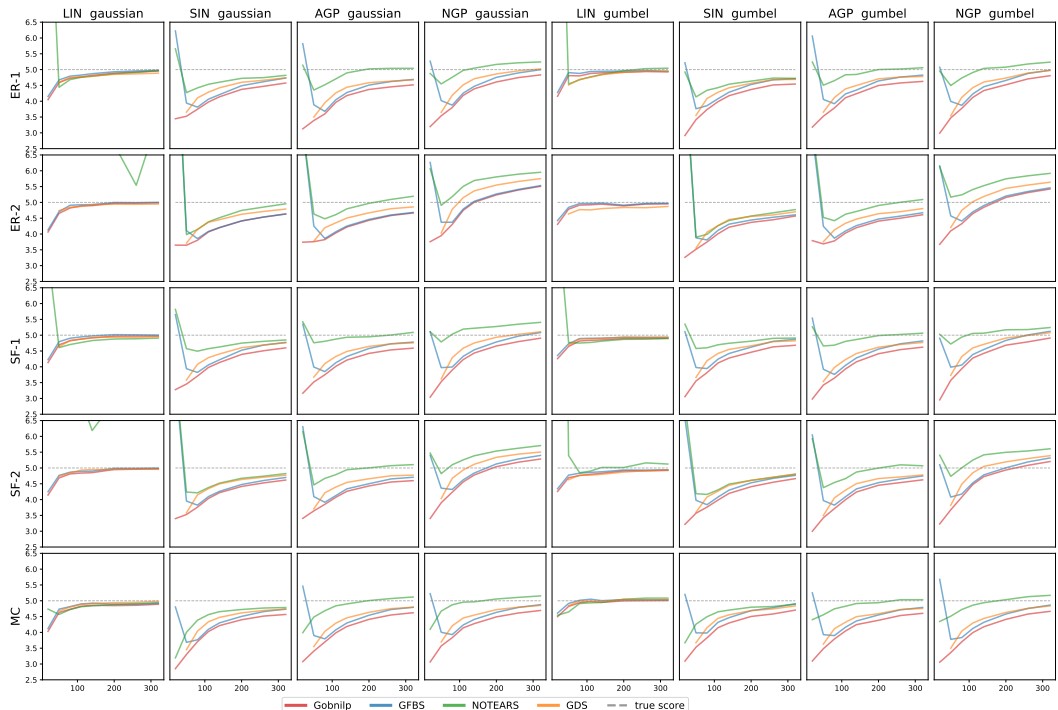

Figure 3: Score of output DAG vs. sample size $n$ for GFBS and 3 other algorithms for $\phi_1$ settings. Left four columns: $Z_i$ is Gaussian distribution with variance 1; Right four columns: $Z_i$ is Gumbel distribution with variance 1. The grey dashed line is the score of the true graph.

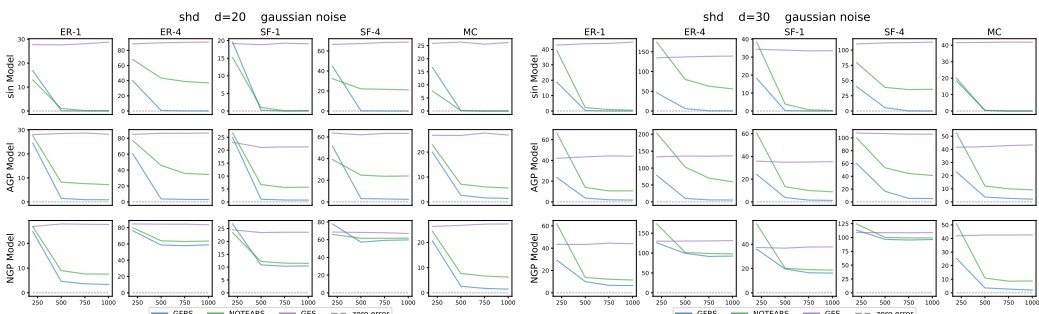

Figure 4: SHD v.s. sample size for $d = 20, 30$ and Gaussian noise.

### H.4.3   Score optimization

We consider the experiment setting under $\phi_1$. Record the estimated score of estimated DAG at each step of iteration (5 for $d = 5$ in total), starting from empty graph. The different sample size is indicated by darkness of the color. The gray dashed line is the true score (5 for $d = 5$).

- Figure 7: Score v.s. iteration for $d = 5$ and Gaussian noise
- Figure 8: Score v.s. iteration for $d = 5$ and t noise
- Figure 9: Score v.s. iteration for $d = 5$ and Gumbel noise

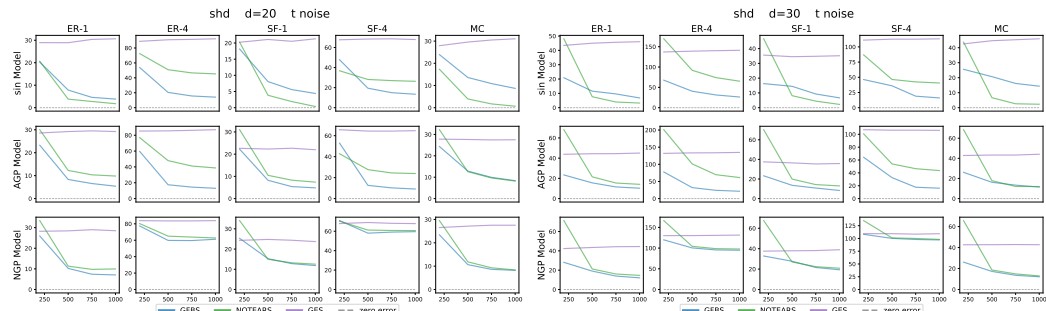

Figure 5: SHD v.s. sample size for $d = 20, 30$ and t noise.

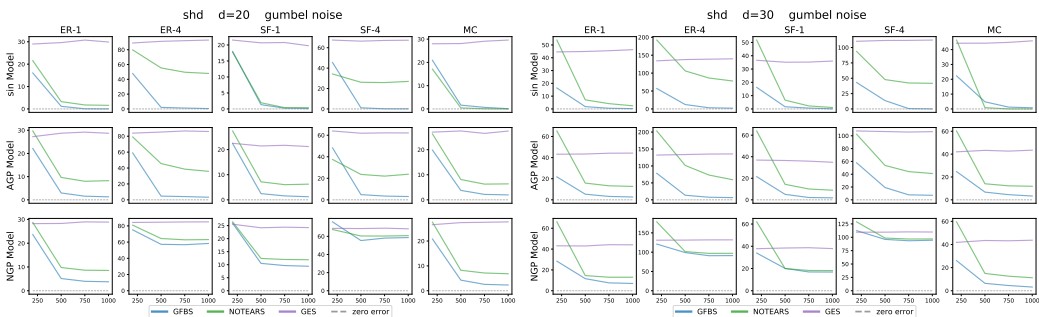

Figure 6: SHD v.s. sample size for $d = 20, 30$ and Gumbel noise.

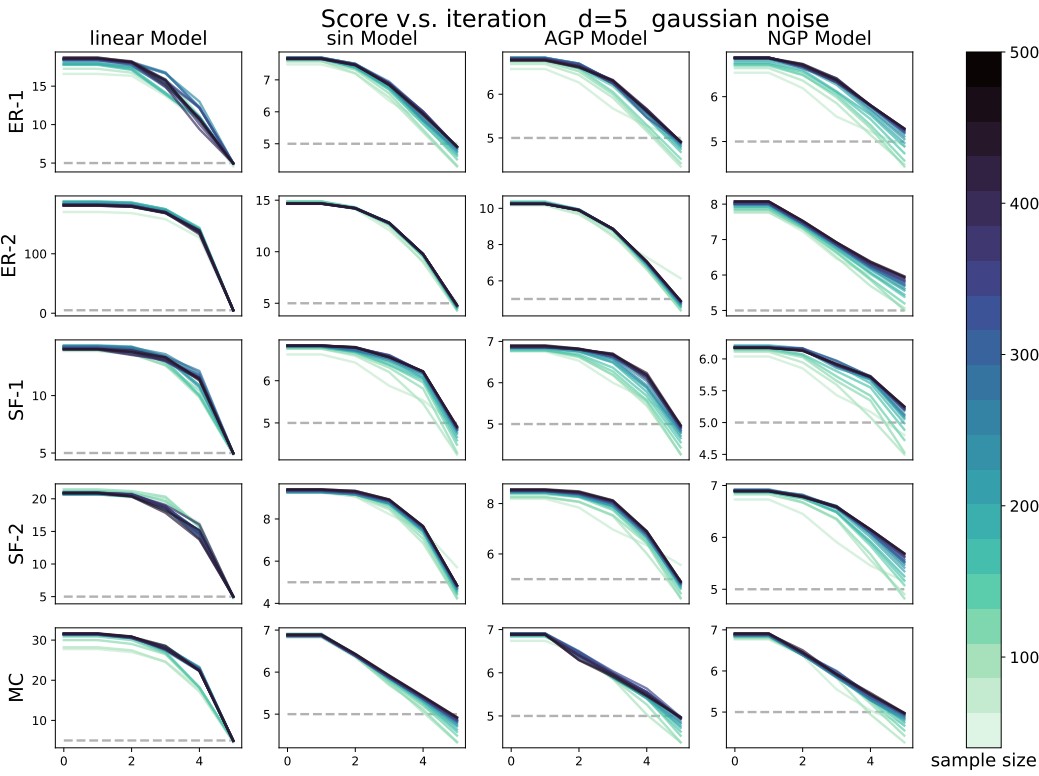

Figure 7: Score v.s. iteration for $d = 5$ and Gaussian noise

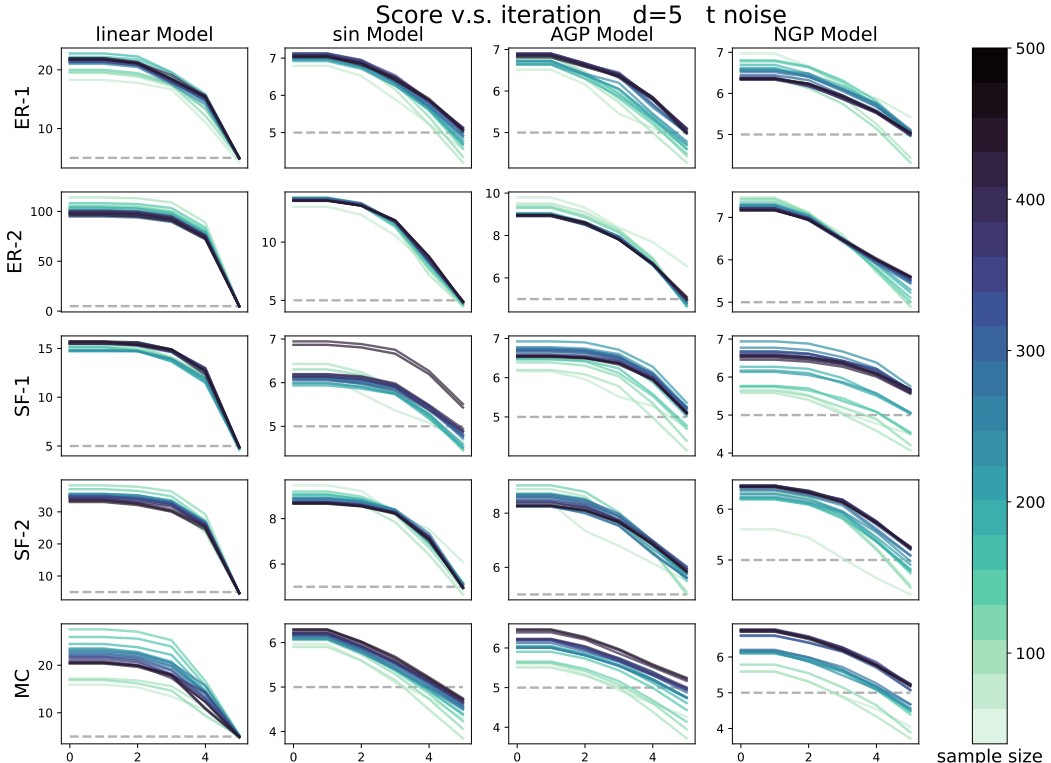

Figure 8: Score v.s. iteration for $d = 5$ and t noise

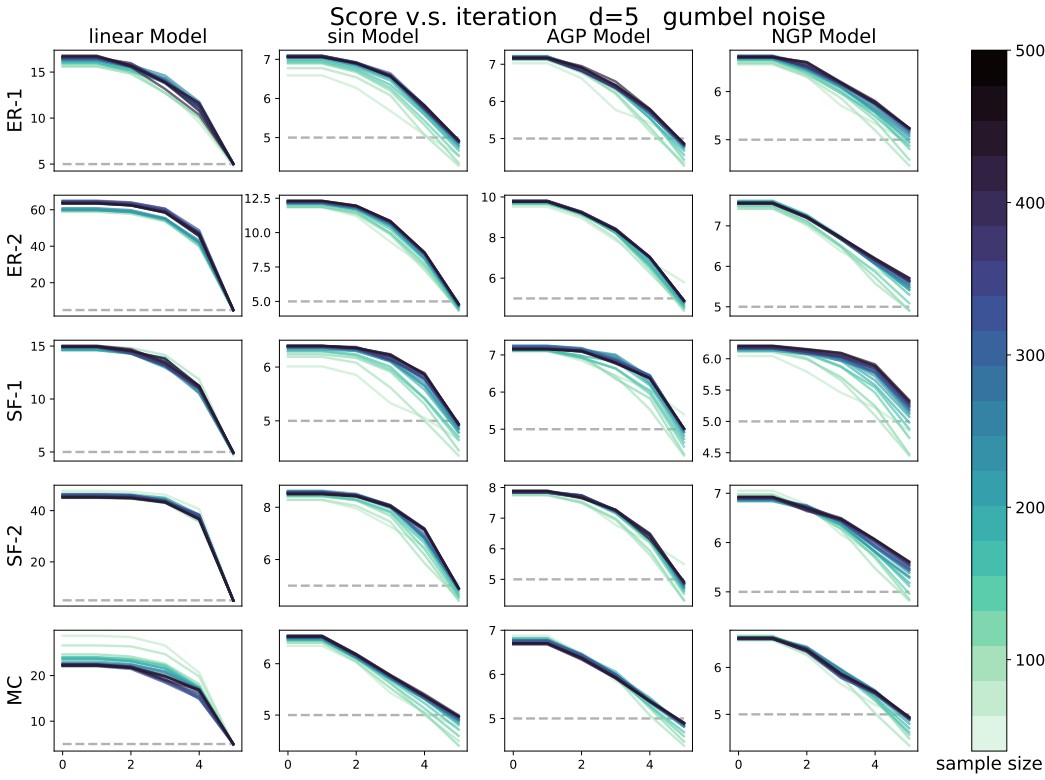

Figure 9: Score v.s. iteration for $d = 5$ and Gumbel noise