# OpenReview forum: "Structure learning in polynomial time: Greedy algorithms, Bregman information, and exponential families"
_NeurIPS.cc/2021/Conference — NeurIPS 2021 Poster_

### Official Review · Reviewer_xMq4 · 2021-07-08

**Rating:** 5
**Confidence:** 3

**Summary:**

This paper studies polynomial-time algorithm of DAG learning, which aims to build a connection between order-based greedy algorithm and score-based methods. A greedy algorithm that sequentially finds an order of the nodes according to a score function is proposed, which is shown to find the optimal score function under proper models. The method on equal-variance Gaussian linear model is generalized using Bregman divergence and Bregman information. Under two key assumptions, properties of the greedy algorithm are proved, which hold in a similar fashion to the variance minimization in gaussian linear model, but in a more general sense. The validity and applicability of the proposed method to several types of score functions and underlying models are evaluated.

**Limitations And Societal Impact:**

Yes.

**Main Review:**

Originality & significance:

This paper finds connection between greedy order search algorithm and score minimization algorithms under particular model assumptions. Most of the results are generalization of the greedy algorithm for equal-variance linear DAG in [Chen et al., 2019], where the gap in variance is generalized to the gap in Bregman score. From the theoretical perspective, the results are not surprising, as they are in the similar sense to [Chen et al., 2019]. However the paper provides a more general perspective in understanding what kinds of DAG models are polynomial-time learnable.

Quality / clarity:

The paper is clearly written and organized.
The algorithm is clearly presented, followed by an explanation to its connection to existing methods in literature.
The main part of Bregman information and score minimization is clearly stated. The theories are correct.
The numerical experiments are illustrative, which shows the performance of the algorithm in some examples.

Questions:

1. It's shown that the forward pass of the algorithm is equivalent to [Chen et al., 2019] under linear gaussian DAG. The backward step is new compared to [Chen et al., 2019] but its role in the algorithm is not explained. I think it is used to eliminated unnecessary edges so that $\hat{W}=W$ with high probability, which could be elaborated a bit.
2. The identification relies on a gap condition ($\Delta$), but the parameter in the backward step also relies on the possibly unknown constant. Is there a way to mitigate this problem?
3. Often the DAG problem is concerned with high dimensionality. The constant gap condition $\Delta$ might not be enough for characterizing the high dimensional case when signal strength might be weak. Is it possible to have more specific considerations in the high dimensional setting?



Reference:
[Chen et al., 2019] W. Chen, M. Drton, and Y. S. Wang. On causal discovery with an equal-variance assumption. Biometrika, 106(4):973–980, 09 2019. ISSN 0006-3444. doi: 10.1093/biomet/asz049.

**Time Spent Reviewing:**

4

---

> ### Author Response · Authors · 2021-08-10
> **Response to Reviewer xMq4**
>
> We thank the reviewer for their feedback. We will address their questions below.
>
> > _the results are not surprising, as they are in the similar sense to [Chen et al., 2019]_
>
> While there is similarity in the conclusions, our proof technique is almost entirely unrelated to [Chen et al 2019], since it can no longer exploit the law of total variance or the covariance structure of linear models. Indeed, [Chen et al, 2019] only analyze the linear case, which is substantially simpler than our nonparametric setting. Finally, the connection with score-based learning is completely absent from this previous work.
>
> > _It's shown that the forward pass of the algorithm is equivalent to [Chen et al., 2019] under linear gaussian DAG. The backward step is new compared to [Chen et al., 2019] but its role in the algorithm is not explained. I think it is used to eliminated unnecessary edges so that \hat{W}=W with high probability, which could be elaborated a bit._
>
> The reviewer’s interpretation is correct, and we will be happy to clarify this. We note that the backward phase is standard in greedy optimization, e.g. Greedy Equivalence Search (GES) [2] or other more general greedy algorithms (see L18-25).
>
> > _The identification relies on a gap condition (Δ), but the parameter in the backward step also relies on the possibly unknown constant. Is there a way to mitigate this problem?_
>
> Thanks for mentioning this practical issue. When the true gap $\Delta$ is unknown, one may choose $\gamma \asymp \sqrt{d\delta_n}$. Then the proof still goes through, and the $\Delta$ term in the sample complexity would be removed, at a cost of replacing $\epsilon$ by $\epsilon^2$.
>
> > _Often the DAG problem is concerned with high dimensionality. The constant gap condition Δ might not be enough for characterizing the high dimensional case when signal strength might be weak. Is it possible to have more specific considerations in the high dimensional setting?_
>
> This is a good point, it’s indeed possible to adapt our parameters in the high dimensional setting. In fact, we *do not require $\Delta$ to be a constant*. $\Delta$ is allowed to depend on $n, d$, or any other parameter in the problem as long as the inequalities in Appendix F go through. For example, it can have any dependency on the dimension of the graph $d$, e.g. $\Delta \asymp 1/d$. Therefore, the interpretation of the role of this gap is flexible in the derived sample complexity. This is an interesting direction to explore in future works.
>
> **References:**
>
> [1] [Chen et al., 2019] W. Chen, M. Drton, and Y. S. Wang. On causal discovery with an equal-variance assumption. Biometrika, 2019.
>
> [2] D. M. Chickering. Optimal structure identification with greedy search. Journal of Machine Learning Research, 3:507–554, 2003.

---

### Official Review · Reviewer_vCrF · 2021-07-10

**Rating:** 6
**Confidence:** 3

**Summary:**

This paper provides a conceptually simple greedy algorithm for solving the Bayesian Network Structure Learning (BNSL) problem. More precisely, the Greedy Forward-Backward Search (BFBS) algorithm is first presented for arbitrary scoring functions. Then, the paper results are refined using different scoring functions, inspired from variance measures (Section 3.1) and Bregman divergences (Sections 4 & 5). Comparative experiments are presented in Section 6.


**Ethical Concerns:**

Not Applicable.

**Limitations And Societal Impact:**

Broader impacts are briefly discussed at the end of Page 9. The (technical) limitations of the approach (in particular the GFBS algorithm) have not really been discussed.

**Main Review:**

Unless I missed something, this paper has established that P = NP:
* The output of GFBS is a minimizer of Problem (1), which is to find a directed acyclic graph that minimizes a scoring function $S$.
* As stated in Lines 149-150, “The running time of GFBS is polynomial in $d$ and the time needed to compute the score $S(.)$.”
* Suppose that $S$ is the BDe metric with the standard assumptions considered in Chickering (1996). For any DAG $W$, $S(W)$ can be computed in time polynomial in $d$ and the size of the dataset $X$. Since the GFBS problem with the BDe metric is NP-hard, this immediately implies that P = NP.
* Alternatively, suppose that $S$ is linear, that is, for any input digraph $W$, $S(W) = \sum_{v \in V} S_v(Q_v)$, where $Q_v$ is the set of parents of $v$ and $S_v(Q) = \sum_{q \in Q} S_v(\{q})$ for all $Q \subseteq V$. Observe that $S$ can be described using a representation that is polynomial in $d$.  Then, for this additive score function, a reduction of BNSL to the Feedback Arc Set problem can be easily found, which again implies that GFBS is solving in polynomial time an NP-hard problem.

Based on this evidence, I would suggest a strong revision of the paper. The correctness of Algorithm 1 should be checked and formally proved. If the output is not a minimizer of (1) but an approximation of it (depending on the scoring function and possibly other parameters), approximation bounds should be found. Note that the aforementioned Feedback Arc Set problem is already APX-hard.

--- After Rebuttal ---

I was originally confused by the fact that the output of their GFBS algorithm (see the third line of Page 4):

$ \mathrm{argmin}_{\mathrm{DAG} W} S(W) $

is a minimizer of Problem (1). Based on the authors’ response, GFBS is optimal (on expectation) for specific classes of cost functions that depend on some properties of the underlying data distribution (actually, $S$ should depend on $\mathcal D$).



**Time Spent Reviewing:**

3

---

> ### Author Response · Authors · 2021-08-10
> **Response to Reviewer vCrF**
>
> It is _not_ the case that “the output of GFBS is a minimizer of Problem (1)”, nor do we claim this, and as such **the first line of the reviewer’s argument does not go through**. We cannot find anywhere in the paper where this claim is made explicitly, although admittedly some of the writing may have been vague on this point. We will carefully revise the wording to ensure there can be no confusion about this. We would also appreciate specific pointers from the reviewer as to where this confusion arose.
>
> (For emphasis, it is perhaps worth highlighting that we have presented **no theoretical results for the BDe score**, precisely for this reason: It is known that optimizing BDe without further constraints is NP-hard.)
>
> You are right that any polynomial time algorithm that exactly minimizes scores such as BDe will prove P = NP. But we **do not** claim in this work that our algorithm exactly minimizes the score. We also state this precisely in L116 in the text: “We do not prove that this algorithm exactly solves (1)”. We also discuss the NP-hardness of this general problem in L59-66 and point out what’s achievable via score-based learning, which forms the foundation for our research.
>
> We apologize if any other sentences in our text made you think we are exactly optimizing the score, we will carefully proofread and make precise any sentences that could be interpreted differently (we are also happy to accept suggestions from the reviewer). We also note that no other reviewers were confused by this. In fact, another reviewer explicitly commented on this: “when polynomial-time recoverability is not precluded by hardness results, the proposed greedy algorithm may also be able to minimize the score” -- this is precisely how we mean our results to be interpreted.
>
> We agree that approximation bounds present an interesting future direction, however, this is not the objective of the current paper, which is more statistical in nature. We prove rigorously that GFBS will output the correct DAG (Theorem 4.6, Corollary 4.7, Theorem 5.4, Theorem 5.6). This is similar to existing results for the popular Greedy Equivalence Search (GES) [2]: In GES, there is no guarantee that an exact optimal solution to (1) is found, but there is a guarantee that the true DAG will be returned [2] (assuming faithfulness, which we stress we do not assume). Perhaps surprisingly, it turns out that exactly optimizing the score and returning the true DAG are different problems (see Example 1 in [1]). Part of the purpose of the present work is to study this further, and to inspire future work to understand this relationship more rigorously (e.g. by obtaining approximation bounds as the reviewer suggests).
>
> Finally, we agree that adding a more detailed discussion of limitations is worthwhile, and will be happy to do so with the extra space in the final version. In particular, a discussion of the P vs NP aspects, and the fact that it is an open problem whether or not GFBS exactly optimizes Bregman scores, will be discussed more carefully.
>
> **References:**
>
> [1] Loh, Po-Ling, and Peter Bühlmann. "High-dimensional learning of linear causal networks via inverse covariance estimation." The Journal of Machine Learning Research 15.1 (2014): 3065-3105.
>
> [2] D. M. Chickering. Optimal structure identification with greedy search. Journal of Machine Learning Research, 3:507–554, 2003.

---

> > ### Comment · Reviewer_vCrF · 2021-08-29
> > **Re: Response**
> >
> > Thanks for your detailed response. I was originally confused by the fact that the output of the GFBS algorithm (see the third line of Page 4):
> >
> > $ \mathrm{argmin}_{\mathrm{DAG} W} S(W) $
> >
> > is a minimizer of Problem (1). Based on your response, GFBS is optimal (on expectation) for specific classes of cost functions that depend on some properties of the underlying data distribution (actually, $S$ should depend on $\mathcal D$). Overall, I am still thinking that the paper could be polished:  there are notational issues that might lead to confusion. But I no longer see any major obstacle, so I will raise the score to marginally above the acceptance threshold.

---

> > > ### Author Response · Authors · 2021-08-31
> > > **Thank you for clarifying**
> > >
> > > We appreciate your response and understanding! We will be certain to correct the line in question as we can now see how it lead to your confusion. We also plan to carefully polish the paper with your and the other reviewers' helpful comments in mind.

---

### Official Review · Reviewer_d421 · 2021-07-16

**Rating:** 6
**Confidence:** 2

**Summary:**

This paper studies the structure learning of graphical models over directed acyclic graphs, and proposes a two-pass greedy algorithm (Greedy Forward-Backward Search) which evaluates a score based on Bregman information to learn directed acyclic graphical models in polynomial time under mild statistical assumptions.

**Limitations And Societal Impact:**

The paper has a short paragraph explaining the possible societal impact, which is typical for a theory-oriented paper.


**Main Review:**

- The algorithm is efficient, requires only a quadratic number of score evaluations.
- The algorithm recovers the correct DAG for properly chosen score functions.
- Conceptually, the proposed algorithm generalizes other order-based algorithms for structure learning.
- The paper proves identifiability of graphical models under mild statistical assumptions.
- The paper is well written and easy to follow.


**Time Spent Reviewing:**

3

---

> ### Author Response · Authors · 2021-08-10
> **Response to Reviewer d421**
>
> Thanks for the positive review and for succinctly summarizing our main contributions accurately. If you have any feedback on how the presentation or the results can be improved, we are happy to accept this feedback as well.

---

### Official Review · Reviewer_YN8z · 2021-07-19

**Rating:** 7
**Confidence:** 4

**Summary:**

This paper acts as a review and a generalization of existing greedy learning algorithms for DAGs, through the introduction of a score function. The paper also introduces a new score function based on Bregman information and provide sample and computational complexities for this function. They provide empirical results for this greedy algorithm in learning models that violate their main assumption, Assumption 4.4.

**Limitations And Societal Impact:**

Yes

**Main Review:**

The strong point of this paper is that it acts a review of existing greedy techniques for learning DAGs and generalizes some of these techniques into their GFBS algorithm. I think the work on determining the sample and computational complexity is worth publishing.

However, I think the authors have to include more substantial discussions in two main areas:

1) How does the sample and computational complexity of using Bregman information as a score compare to that of other greedy techniques that fall into a specific case of the GFBS algorithm? I think this is an important area to address and if the Bregman information does improve upon the sample and computational complexity of the greedy algorithm then this enhances the introduction of a general function significantly.

2) More discussion needs to be included for the experimental results on how these models used for the experiments violate Assumption 4.4.

3) In the section on main contributions, the authors describe their learning algorithm is vertex based, whereas existing algorithms are edge based. In the next point, they mentioned that some existing algorithms fall under the special case of their function by determining a score functions. I think these two points sound like they are at odds which each other since if the second point is true, then it needs to be discussed how many of existing algorithms are edged based or vertex based. Also, the advantages of vertex based algorithms need to be discussed.

**Time Spent Reviewing:**

9

---

> ### Author Response · Authors · 2021-08-10
> **Response to Reviewer YN8z**
>
> We thank the reviewer for the comments. We address the reviewer’s concerns below and will  also incorporate some of these clarifications into the main text.
>
> > _How does the sample and computational complexity of using Bregman information as a score compare to that of other greedy techniques that fall into a specific case of the GFBS algorithm?_
>
> This is a great question. In our view, the advantage of our approach is to provide new sample complexity results where previously _no sample complexities were known period_. This is the advantage of the exponential family and Bregman information approach in Section 4. For example, our theoretical results in Section 5 apply to these general scores, which significantly extend previous work for the least squares loss. We also emphasize that these general score functions are highly relevant in ML as they are directly related to the optimization of the log-likelihood (we point this out in L228-231).
>
> For the least squares loss, our results unify and match existing results. In the linear case, EqVar [1,2] is a special case of GFBS with the same complexity result. For nonlinear models, very little is known about finite sample complexity. One exception is the work [3], which is also a special case of GFBS, and applying our results for least squares loss (i.e. $\phi(x) = x^2$), we match the sample complexity of [3]. For non-LS loss with non-Gaussian exponential family models, we are not aware of any finite-sample results prior to our work.
>
> > _More discussion needs to be included for the experimental results on how these models used for the experiments violate Assumption 4.4_
>
> We apologize for any confusion here, as these details have been relegated to Appendix G. Essentially, Assumption G.1 relaxes the Assumption 4.4, and we provide an example together with experiments to demonstrate it. Both (a) non-violation and (b) violation situations are considered (at L764). We are happy to discuss more about it if there is anything unclear, and will be happy to add a bit more detail to the main paper on this in the final version.
>
> > _In the section on main contributions, the authors describe their learning algorithm is vertex based, whereas existing algorithms are edge based. In the next point, they mentioned that some existing algorithms fall under the special case of their function by determining a score functions. I think these two points sound like they are at odds which each other since if the second point is true, then it needs to be discussed how many of existing algorithms are edged based or vertex based. Also, the advantages of vertex based algorithms need to be discussed._
>
> We apologize for the confusing wording on our part, and thank the reviewer for pointing this out. To clarify: Existing _score-based_ greedy algorithms (such as GES or hill climbing) are edge based, whereas recent _order-based_ algorithms (such as [1,2,3]) are vertex-based. Our point is that GFBS shows that [1,2,3] are in fact _score-based_ greedy algorithms, but in contrast to existing edge-based approaches, they are vertex-based. We accept that this distinction is a bit technical and the current writing is not clear on this, and we will be sure to clarify this carefully in the final version.
> Regarding advantages, the main advantages would be that the vertex-based approach leads to provably poly-time algorithms (as in our work), vs. edge-based approaches which are worst-case exponential (such as GES).
>
> **References:**
>
> [1] A. Ghoshal and J. Honorio. Learning identifiable gaussian bayesian networks in polynomial time and sample complexity. In Advances in Neural Information Processing Systems 30, pages 376 6457–6466. 2017.
>
> [2] Chen, Wenyu, Mathias Drton, and Y. Samuel Wang. "On causal discovery with an equal-variance assumption." Biometrika 106.4 (2019): 973-980.
>
> [3] M. Gao, Y. Ding, and B. Aragam. A polynomial-time algorithm for learning nonparametric causal graphs. In Advances in Neural Information Processing Systems, 2020.

---

> ### Comment · Reviewer_YN8z · 2021-09-01
> **Thank you for the response.**
>
> I feel that the authors have addressed some of my concerns, I am willing to increase my score.

---

### Official Review · Reviewer_tJvN · 2021-07-20

**Rating:** 7
**Confidence:** 4

**Summary:**

The paper proposes a score-based algorithm to recover the structure of a Bayesian network. The algorithm first constructs a topological order, by greedily adding the vertex that contributes the least to the score, with all ancestors as its parents. The algorithm then prunes the edges by thresholding the contribution of the edge relative to its absence. The overall runtime is a polynomial (in the size of the graph), times the cost of calculating a score. Theoretical results are proven generally for scores given by Bregman divergences. Under a set of assumptions, for the population case the recovery is exact and for the sample case, with a modification of the pruning stage, the recovery holds with high probability. The work is also supported by experiments.

**Ethical Concerns:**

No ethical concerns.

**Limitations And Societal Impact:**

Authors adequately discuss some limitations and impact.

**Main Review:**

### Strengths

+ The problem of Bayesian network structure learning continues to be relevant, particularly invigorated by causal models. The polynomial complexity of the proposed algorithm and its relationship with topological order based algorithms make this a contribution worth sharing with the community. _(originality, significance)_

+ The paper is well-written, the motivation, contribution, and technical details are clear. _(clarity)_

### Weaknesses

- The Bregman divergence and exponential family formulation in Section 4.1, including Lemma 4.2, is known. _(novelty)_

### Suggestions

* In the algorithm, during the first stage $W$ needs to be updated too for the notation to make sense. The way I understand it, $W$ becomes $W[T\to i]$, before $T$ gets updated.

* The notation used in Definition 5.5 is following that of the proof of Theorem 5.4, but this doesn’t read well in the flow of the main text. It would be better rewritten or at least if that notation is introduced. Clearly comparing also to the original second (pruning) stage would help.

* Questions: In the identifiable case, the context of Theorem 4.6 and Corollary 4.7, I may be confused, but doesn’t the recovery of the true model imply that the score is indeed minimized? (Granted that this is under assumptions.) Is the conjecture that in more general settings, when polynomial-time recoverability is not precluded by hardness results, the proposed greedy algorithm may also be able to minimize the score? What $\Delta$ (or $\gamma$, pruning threshold) was used in the experiments, and what can one do when this is not known? How do you explain the large gap for the (SF-2, NGP) experiment?

* Typos: `line 140` _Furthermore_, `line 313` _its parents_, references [32] and [33] repeat

[Edit: Thank you authors for your clarifying response.]

**Time Spent Reviewing:**

4.5

---

> ### Author Response · Authors · 2021-08-10
> **Response to Reviewer tJvN**
>
> We thank the reviewer for their constructive comments and for helpful suggestions to improve the text. Although the relationship between Bregman divergences and exponential families is indeed known (L224-225, L231-232, L622-623), a contribution of our work is to highlight that this relationship can be further exploited in the context of Bayesian Network learning. To the best of our knowledge, this connection between greedy algorithms, Bregman divergences, and Bayesian Network learning has not been observed in the literature before.
>
> Below we respond to some of the specific questions:
>
> > _W becomes W[T→i], before T gets updated._
>
> This is correct, thanks for catching that.
>
> > _The notation used in Definition 5.5 is following that of the proof of Theorem 5.4, but this doesn’t read well in the flow of the main text. It would be better rewritten or at least if that notation is introduced. Clearly comparing also to the original second (pruning) stage would help._
>
> That’s a very helpful comment, we will clarify these in the final version.
>
> > _doesn’t the recovery of the true model imply that the score is indeed minimized?(Granted that this is under assumptions.)_
>
> Although this is usually expected to be the case, in the case of DAG learning, surprisingly this may not be the case. This is discussed in detail in [1]; see e.g. their Example 1. Score based algorithms in general attempt to learn the true model by way of minimizing the score but it’s possible that the graph which minimizes the score could be different from the true model. This is a subtle point that we should have emphasized, and we will plan to add a remark to this effect in the final version.
>
> > _Is the conjecture that in more general settings, when polynomial-time recoverability is not precluded by hardness results, the proposed greedy algorithm may also be able to minimize the score?_
>
> Precisely. An important direction for future research is to analyze in which settings this conjecture will hold. Our discussion at L59-66 was admittedly a bit terse on this point, so we will plan to expand and update this discussion in the final version.
>
> > _What Δ (or γ, pruning threshold) was used in the experiments, and what can one do when this is not known?_
>
> In Appendix H.2 (L812), we point out the threshold used in the experiments, $\gamma=0.05$ for $\phi_1$ and $\gamma=0.0005$ for $\phi_2$. When the true $\Delta$ is unknown, one may choose $\gamma \asymp \sqrt{d\delta_n}$.  The proof still goes through, and the $\Delta$ term in the sample complexity would be removed, with a cost of replacing $\epsilon$ by $\epsilon^2$.
>
> > _How do you explain the large gap for the (SF-2, NGP) experiment?_
>
> This is the hardest setting we have tested. For SF-2 graphs, the indegree can be large for some particular nodes, and NGP is the most general and most difficult nonlinear function to estimate. Thus with the same sample size as other settings, the performance may be slightly worse. But the gap will be gone as soon as sample size increases.
>
> > _Typos: line 140 Furthermore, line 313 its parents, references [32] and [33] repeat_
>
> Thank you, we will fix this.
>
> **References:**
>
> [1] Loh, Po-Ling, and Peter Bühlmann. "High-dimensional learning of linear causal networks via inverse covariance estimation." The Journal of Machine Learning Research 15.1 (2014): 3065-3105.

---

### Decision · Program_Chairs · 2021-09-28

**Decision:**

Accept (Poster)

**Comment:**

This paper proposes a general greedy algorithm for score-based learning of Bayesian networks from data. The algorithm generalizes a number of prior approaches to learning such models by using a score function that is based on a sum of Bregman informations.

The algorithm is simple and computationally efficient (assuming that the score function can be computed efficiently). It is shown to succeed under what seem like fairly restrictive assumptions which generalize to the Bregman setting the corresponding restrictive assumptions used in some prior papers.

Thus the main strength of the paper is in its generality and unification of prior algorithms.

The main weakness are that (1) given the restrictiveness of the assumed conditions, it would have been nice to have obtained some insight about whether these are necessary, and (2) the main contribution is in placing prior ideas in a more general setting. Thus, it is not clear what substantial new insights into the basic phenomena the current work provides. Definitively addressing either of these weaknesses is a worthwhile direction for future work.

**Consistency Experiment:**

NeurIPS has a long history of experimentation. In 2014, NeurIPS ran an experiment in which 10% of submissions were reviewed by two independent committees to quantify the randomness in the review process. This year, we repeated a variant of this experiment to see how the quality of the review process has changed over time.  This paper was part of the experiment and was therefore assigned to two committees (consisting of reviewers, an Area Chair, and a Senior Area Chair) that reached independent decisions.  If both committees made the same recommendation, this recommendation was followed. If a single committee recommended acceptance, the paper was accepted (with the exception of a few cases in which the other committee identified what we considered a fatal flaw, e.g., an error in a key result).

This copy’s committee reached the following decision: **Accept (Poster)**

The other committee assigned to the paper recommended **Reject**.  You can find the other set of reviews, along with any follow up discussion with the authors here:
https://openreview.net/forum?id=Fv0DPhwB6o9